# Learning to express reward prediction error-like dopaminergic activity requires plastic representations of time

Ian Cone [1,2,3], Claudia Clopath[1] & Harel Z. Shouval [2,4] ✉

The dominant theoretical framework to account for reinforcement learning in the brain is temporal difference learning (TD) learning, whereby certain units signal reward prediction errors (RPE). The TD algorithm has been traditionally mapped onto the dopaminergic system, as firing properties of dopamine neurons can resemble RPEs. However, certain predictions of TD learning are inconsistent with experimental results, and previous implementations of the algorithm have made unscalable assumptions regarding stimulus-specific fixed temporal bases. We propose an alternate framework to describe dopamine signaling in the brain, FLEX (**F**lexibly **L**earned **E**rrors in E**x**pected Reward). In FLEX, dopamine release is similar, but not identical to RPE, leading to predictions that contrast to those of TD. While FLEX itself is a general theoretical framework, we describe a specific, biophysically plausible implementation, the results of which are consistent with a preponderance of both existing and reanalyzed experimental data.

The term reinforcement learning (RL) is used in machine learning[1], behavioral science[2], and neurobiology[3], to denote learning on the basis of rewards or punishment. RL is necessary for animals and humans to learn how to obtain rewards such as food and to avoid dangers, such as predators[2]. It is also necessary in order to learn efficient spatial navigation towards areas where there are rewards and away from areas that are dangerous[4].

The positive and negative reinforcers are often delayed in time with respect to the time when decisions are made. For example, one might need to turn right on Main in order to get to a restaurant that is on Broad Street, but only arrives at the restaurant after a few more turns and a few more minutes. Still, it is necessary to associate the future reward with the specific decision to turn right on Main Street. Although experiments show that the brain is able to associate a cue or action with a delayed reinforcement signal[5,6], the identity of the biological mechanism tracking these temporal delays remains unresolved.

One type of RL algorithm is temporal difference (TD) learning, which was designed for machine learning purposes[7,8]. It has the normative goal of estimating future rewards when rewards can be delayed in time with respect to the actions or cues that engendered these rewards[1].

One of the variables in TD algorithms is called reward prediction error (RPE), which is the difference between the discounted predicted reward at the current state and the discounted predicted reward plus the actual reward at the next state. TD learning theory gained traction in neuroscience once it was demonstrated that firing patterns of dopaminergic neurons in the ventral tegmental area (VTA) during reinforcement learning resemble RPE[5,9,10].

Implementations of TD using computer algorithms are straightforward, but are more complex when they are mapped onto plausible neural machinery[11–13]. Current implementations of neural TD assume a set of temporal basis-functions[13,14], which are activated by external cues. For this assumption to hold, each possible external cue must activate a separate set of basis-functions, and these basis-functions must tile all possible learnable intervals between stimulus and reward.

In this paper, we argue that these assumptions are unscalable and therefore implausible from a fundamental conceptual level, and demonstrate that some predictions of such algorithms are

[1]Department of Bioengineering, Imperial College London, London, UK. [2]Department of Neurobiology and Anatomy, University of Texas Medical School at Houston, Houston, TX, USA. [3]Applied Physics Program, Rice University, Houston, TX, USA. [4]Department of Electrical and Computer Engineering, Rice University, Houston, TX, USA. ✉e-mail: harel.shouval@uth.tmc.edu

inconsistent with various established experimental results. Instead, we propose that temporal basis functions used by the brain are themselves learned. We call this theoretical framework: **F**lexibly **L**earned **E**rrors in e**X**pected Reward, or FLEX for short. We also propose a biophysically plausible implementation of FLEX, as a proof-of-concept model. We show that key predictions of this model are consistent with actual experimental results but are inconsistent with some key predictions of the TD theory.

## Results

### TD learning with a fixed feature-specific temporal-basis

The original TD learning algorithms assumed that agents can be in a set of discrete labeled states (s) that are stored in memory. The goal of TD is to learn a value function such that each state becomes associated with a unique value ($V(s)$) that estimates future discounted rewards. Learning is driven by the difference between values at two subsequent states, and hence such algorithms are called temporal difference algorithms. Mathematically this is captured by the update algorithm: $V(s) \leftarrow V(s) + \alpha(r(s') + \gamma V(s') - V(s))$, where $s'$ is the next state and $r(s')$ is the reward in the next state, $\gamma$ is an optional discount factor and $\alpha$ is the learning rate.

The term in the brackets in the right-hand side of the equation is called the RPE. It represents the difference between the estimated value at the current state and the estimated discounted value at the next state in addition to the actual reward at the next state. If RPE is zero for every state, the value function no longer changes, and learning reaches a stable state. In experiments that link RPE to the firing patterns of dopaminergic neurons in VTA, a transient conditioned stimulus (CS) is presented to a naïve animal followed by a delayed reward (also called unconditioned stimulus or US, Fig. 1a). It was found that VTA neurons initially respond at the time of reward, but once the association between stimulus and reward is learned, dopaminergic neurons stop firing at the time of the reward and start firing at the time of the stimulus (Fig. 1b). This response pattern is what one would expect from TD learning if VTA neurons represent RPE[5].

Learning algorithms similar to TD have been very successful in machine learning[8,15]. In such implementations, the state (s) could, for example, represent the state of the chess board, or the coordinates in space in a navigational task. Each of these states could be associated with a value. The state space in such examples might be very large, but the values of all these different states could be feasibly stored in a computer's memory. In some cases, a similar formulation seems feasible for a biological system as well. For example, consider a 2-D navigation problem, where each state is a location in space. One could imagine that each state would be represented by the set of hippocampal place cells activated in this location[16], and that another set of neurons would encode the value function, while a third population of neurons (the "RPE") neurons would compare the value at the current and subsequent state. On its face, this seems to be a reasonable assumption.

However, in contrast to cases where a discrete set of states might have a straightforward biological implementation, there are many cases in which this machine learning-inspired algorithm cannot be implemented simply in biological machinery. For example, in experiments where reward is delivered with a temporal delay with respect to the stimulus offset (Fig. 1), an additional assumption of a preexisting temporal basis is required[12].

### Is it plausible to assume a fixed temporal-basis in the brain for every possible stimulus?

Consider the simple canonical example of Fig. 1. In the time interval between the stimulus and the reward, the animal does not change its location, nor does its sensory environment change in any predictable manner. The only thing that changes consistently within this interval is time itself. Hence, in order to mark the states between stimulus and reward, the brain must have some internal representation of time, an

internal clock that tracks the time since the start of the stimulus. Note however that before the conditioning starts, the animal has no way of knowing that this specific sensory stimulus has unique significance and therefore each specific stimulus must a priori be assigned its own specific temporal representation.

This is the main hurdle of implementing TD in a biophysically realistic manner - figuring out how to represent the temporal basis upon which the association between cue and reward occurs (Fig. 1a). Previous attempts were based on the assumption that there is a fixed cue-specific temporal basis, an assumption which has previously been termed "a useful fiction"[12]. The specific implementations include the commonly utilized tapped delay lines[5,10,17] (or the so-called complete serial compound), which are neurons triggered by the sensory cue, but that are active only at a specific delay, or alternatively, a set of cue-specific neuronal responses that are also delayed but have broader temporal support which increases with an increasing delay (the so called "microstimuli")[12,14] (Fig. 1c).

For this class of temporal representations, the delay time between cue and reward is tiled by a chain of neurons, with each neuron representing a cue-specific time (sometimes referred to as a "microstate") (Fig. 1c). In the simple case of the complete serial compound, the temporal basis is simply a set of neurons that have non-overlapping responses that start responding at the cue-relative times: $t_{cue}$, $t_{cue} + dt$, $t_{cue} + 2dt$, …, $t_{reward}$. The learned value function (and in turn the RPE) assigned to a given cue at time t is then given by a learned weighted sum of the activations of these microstates at time t (Fig. 1d).

We argue that the conception of a fixed cue-dependent temporal basis makes assumptions that are difficult to reconcile with biology. First, since one does not know a priori whether presentation of a cue will be followed by a reward, these models assume implicitly that every single environmental cue (or combination of environmental cues) must trigger its own sequence of neural microstates, prior to potential cue-reward pairing (Fig. 1e). Further, since one does not know a priori when presentation of a cue may or may not be followed by a reward, these models also assume that microstate sequences are arbitrarily long to account for all possible (or a range of possible) cue-reward delays (Fig. 1f). Finally, these microstates are assumed to be reliably reactivated upon subsequent presentations of the cue, e.g., a neuron that represents $t_{cue} + 3dt$ must always represent $t_{cue} + 3dt$ - across trials, sessions, days, etc. However, implementation of models that generate a chain-like structure of activity can be fragile to biologically relevant factors such as noise, neural dropout, and neural drift, all of which suggest that the assumption of reliability is problematic as well (Fig. 1g). The totality of these observations implies that on the basis of first principles, it is hard to justify the idea of the fixed feature-specific temporal basis, a mechanism which is required for current supposedly biophysical implementations of TD learning.

Although a fixed set of basis functions for every possible stimulus is problematic, one could assume that it is possible to replace this assumption with a single set of fixed, universal basis-functions. An example of a mechanism that can generate such general basis functions is a fixed recurrent neural network (RNN). Instead of the firing of an individual neuron representing a particular time, here the entire network state can be thought of as a representation of a cue-specific time. This setup is illustrated in Fig. 2a.

To understand the consequences of this setup, we assume a simple environment in which one specific stimulus (denoted as stimulus C) is always followed 1000 ms later by a reward; this stimulus is the CS. However, it is reasonable to assume for the natural world that this stimulus exists among many other stimuli that do not predict a reward. For simplicity we consider 3 stimuli, A, B, and C, which can appear at any possible order, as shown in Fig. 2b, but in which stimulus C always predicts a reward with a delay of 1000 ms.

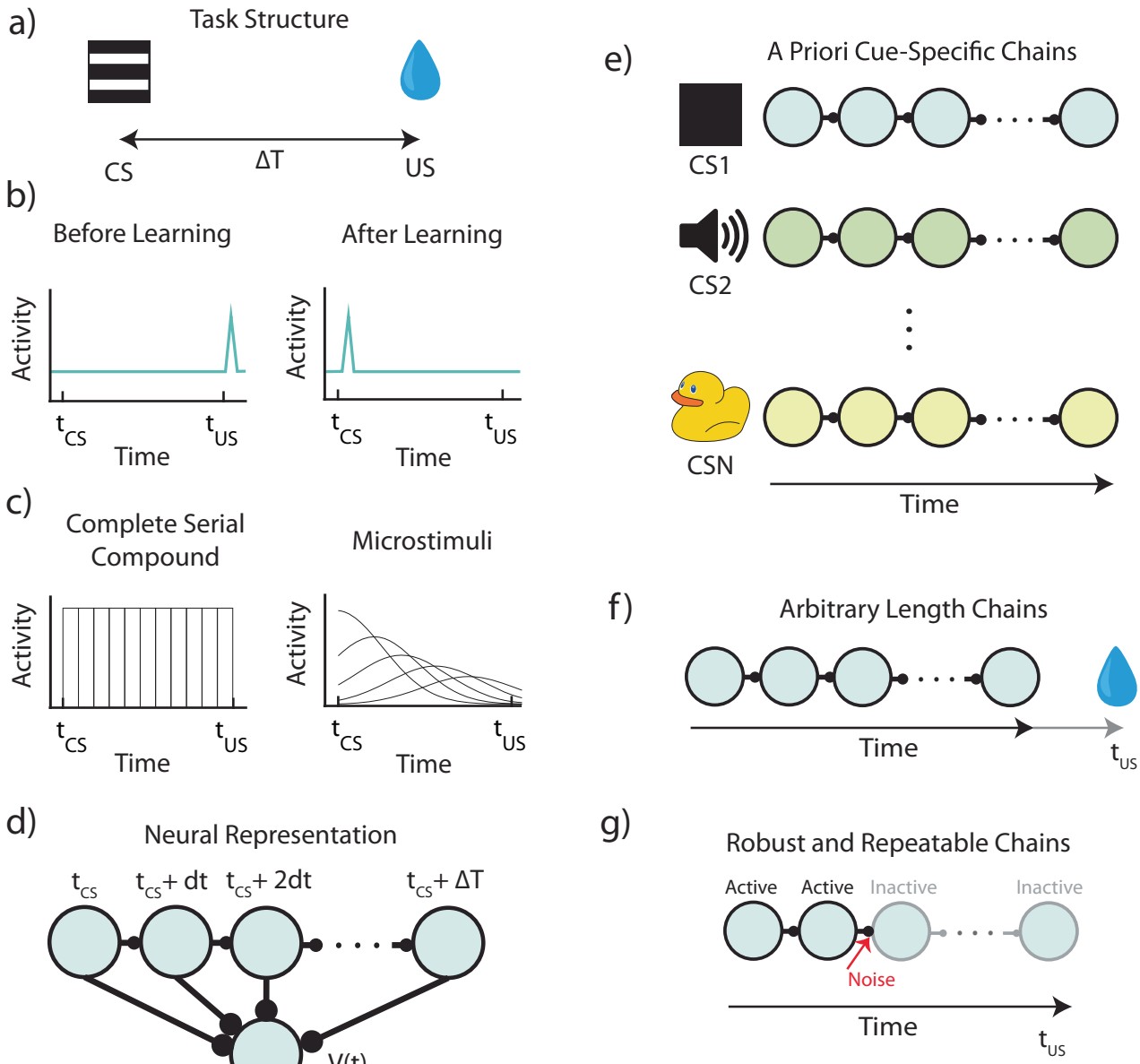

**Fig. 1 | Structure and assumptions of temporal bases for temporal difference learning. a** Diagram of a simple trace conditioning task. A conditioned stimulus (CS) such as a visual grating is paired, after a delay ΔT, with an unconditioned stimulus (US) such as a water reward. **b** According to the canonical view, dopaminergic (DA) neurons in the ventral tegmental area (VTA) respond only to the US before training, and only to the CS after training. **c** In order to represent the delay period, temporal difference (TD) models generally assume neural "microstates" which span the time in between cue and reward. In the simplest case of the complete serial compound (left) the microstimuli do not overlap, and each one uniquely represents a different interval. In general, though (e.g.: microstimuli, right), these microstates can overlap with each other and decay over time. **d** A weighted sum of these microstates determines the learned value function V(t). **e** An agent does not know a priori which cue will subsequently be paired with reward. In turn, microstate TD models implicitly assume that all N unique cues or experiences in an environment each have their own independent chain of microstates before learning. **f** Rewards delivered after the end of a particular cue-specific chain cannot be paired with the cue in question. The chosen length of the chain therefore determines the temporal window of possible associations. **g** Microstate chains are assumed to be reliable and robust, but realistic levels of neural noise, drift, and variability can interrupt their propagation, thereby disrupting their ability to associate cue and reward.

We simulated the responses of such a fixed RNN to different stimulus combinations (see Methods). The complex RNN activity can be viewed as a projection to a subspace spanned by the first two principal components of the data. In Fig. 2c, d we show a projection of the RNN response for two different sequences, A-B-C and B-A-C respectively, aligned to the time of reward. In Fig. 2e we show the two trajectories side by side in the same subspace, starting with the presentation of stimulus C.

Note, that in the RNN (Fig 2c–e), every time stimulus C appears, it generates a different temporal response, depending on the preceding stimuli. These temporal patterns can also be changed by a subsequent stimulus that may appear between the CS and the US. These results mean that such a fixed RNN cannot serve as a universal basis function because its responses are not repeatable.

There are potential workarounds, such as forcing the network states representing the time since stimulus C to be the same across trials. This is equivalent to learning the weights of the network such that all possible "distractor" cues pass through the network's null space. This means that the stimulus resets the network and erases its memory, but that other stimuli have no effect on the network.

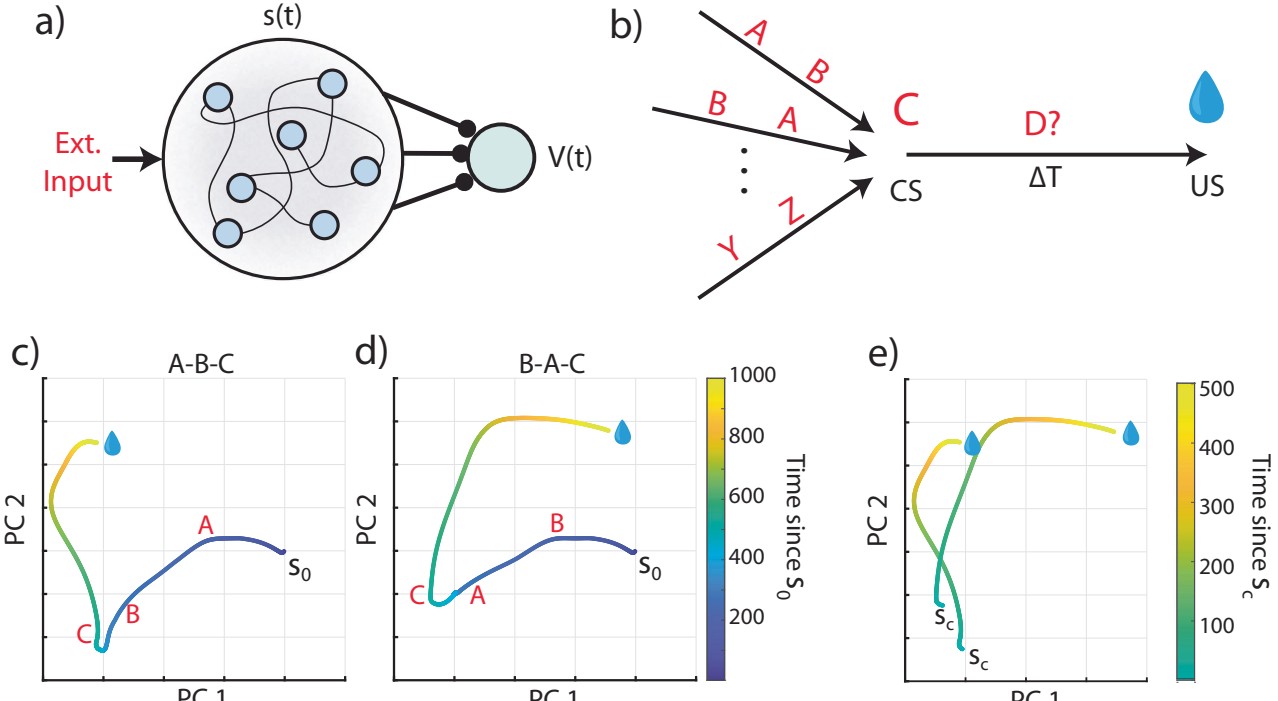

**Fig. 2 | A fixed recurrent neural network as a basis function generator.**
**a** Schematic of a fixed recurrent neural network (RNN) as a temporal basis. The network receives external inputs and generates states s(t), which act as a basis for learning the value function V(t). Compare to Fig. 1c. **b** Schematic of the task protocol. Every presentation of C is followed by a reward at a fixed delay of 1000 ms. However, any combination or sequence of irrelevant stimuli may precede the conditioned stimulus C (they might also come after the CS, e.g. A, C, B). **c** Network activity, plotted along its first two principal components, for a given initial state $s_0$ and a sequence of presented stimuli A-B-C (red letter is displayed at the time of a given stimulus' presentation). **d** Same as **c** but for input sequence B-A-C. **e** Overlay of the A-B-C and B-A-C network trajectories, starting from the state at the time of the presentation of C (state $s_c$). The trajectory of network activity differs in these two cases, so the RNN state does not provide a consistent temporal basis that tracks the time since the presentation of stimulus C.

Generally, one would have to train the RNN to reproduce a given dynamical pattern representing C->reward, while also being invariant to noise or task-irrelevant dimensions, the latter of which can be arbitrarily high and would have to be sampled over during training.

However, this approach requires a priori knowledge that C is the conditioned stimuli (since C-> reward is the dynamical pattern we want to preserve) and that the other stimuli are nuisance stimuli. This leaves us with quite a conundrum. In the prospective view of temporal associations assumed by TD, to learn that C is associated with reward, we require a steady and repeatable labeled temporal basis (i.e., the network tracks the time since stimulus C). However, to train an RNN to robustly produce this basis, we need to have previously learned that C is associated with reward and that the other stimuli are not. As such, these modifications to the RNN, while mathematically convenient, are based on unreasonable assumptions.

Since we are considering the biophysical plausibility of these methods, we omit the consideration of highly connected recurrent networks which use training algorithms such as back propagation through time (BPTT)[18] or FORCE[19,20], which are biophysically implausible as they use credit assignment based on gradients which do not preserve locality/causality. It is indeed possible to learn complex tasks with plastic RNN's without assuming an a priori temporal basis and the resulting neuronal dynamics might resemble cortical dynamics[21]. Such approaches indeed overcome biophysically implausible assumptions about the network structure (i.e., a priori cue-specific chains, Fig. 1e) but do so by using biophysically implausible learning rules.

## Models of TD learning with a fixed temporal-basis show inconsistencies with data
Apart from making problematic assumptions about a pre-existing temporal basis, models of TD learning also make some predictions that

are inconsistent with experimental data. In recent years, several experiments presented evidence of neurons with temporal response profiles that resemble temporal basis functions[22–27], as depicted schematically in Fig. 3a. While there is indeed evidence of sequential activity in the brain spanning the delay between cues and rewards (such as in the striatum and hippocampus), these sequences are generally observed after association learning between a stimulus and a delayed reward[22,25,26]. Some of these experiments have further shown that if the interval between stimulus and reward is extended, the response profiles either remap[25], or stretch to fill out the new extended interval[22], as depicted in Fig. 3a. The fact that these sequences are observed after training and that the temporal response profiles are modified when the interval is changed supports the notion of plastic stimulus-specific basis functions, rather than of a fixed set of basis function for each possible stimulus. Mechanistically, these results suggest that the naïve network might generate a generic temporal response profile to novel stimuli before learning, resulting from the network's initial connectivity.

In the canonical version of TD learning (TD(0)), RPE neurons exhibit a bump of activity that moves back in time from the US to the CS during the learning process (Fig. 3b-left). Subsequent versions of TD learning, called TD(λ), which speed up learning by the use of a memory trace, have a much smaller, or no noticeable moving bump (Fig. 3b, center and right), depending on the length of the memory trace, denoted by λ. Most published experiments have not shown VTA neuron responses during the learning process. In one prominent example by Pan et al. in which VTA neurons are observed over the learning process[28], (depicted schematically in Fig. 2c) no moving bump is observed, prompting the authors to deduce that such memory traces exist. In a more recent paper by Amo et al. a moving bump is reported[29]. In contrast, in another recently published paper, no

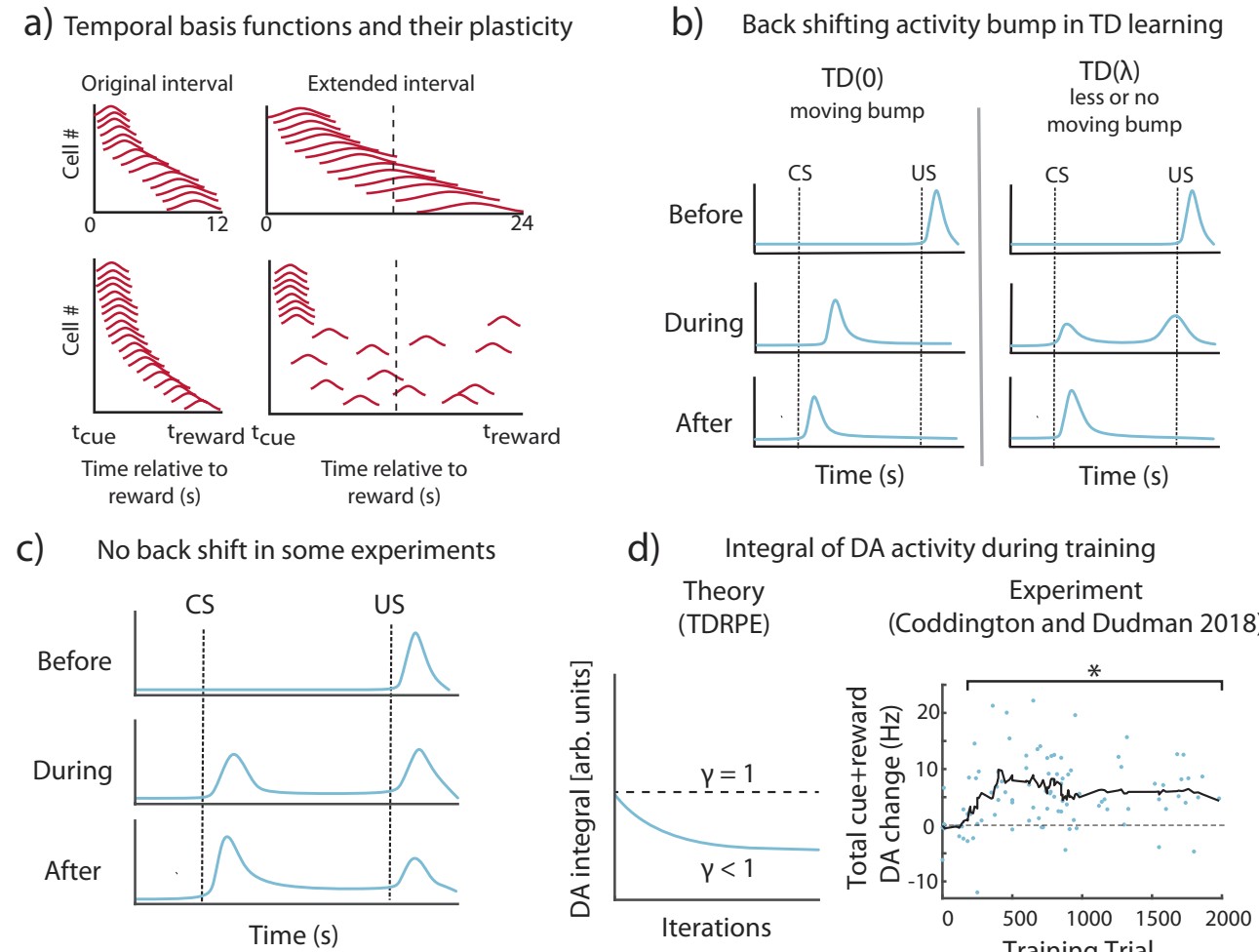

**Fig. 3 | Certain features of experimental results run counter to predictions of TD. a** Putative temporal basis functions (in red) observed in experiments[22,25] develop over training, shown here schematically. If, after training on a given interval between the conditions stimulus (CS) and the unconditioned stimulus (US), the interval is scaled, the basis-functions also change. Recordings in striatum[22] show these basis-functions scale with the modified interval (top), while in recordings from hippocampus[25] (bottom), they are observed to redistribute to fill up the new interval. **b** According to the $TD(0)$ (temporal difference learning with no traces) theory, RPE neuron activity (blue) during learning exhibits a backward moving bump, from the time of the US to the time of the CS (left). For $TD(\lambda)$ (TD with trace decaying at a time constant $\lambda$) the bump no longer appears (right). **c** A schematic depiction of experiments where there is no observation of a backward shifting bump[28,30]. **d** The integral of dopamine neuron (DA) activity according to TD theory

(left) should be constant over training (for $\gamma = 1$, dotted line) or decreasing monotonically for ($\gamma < 1$, blue line). Right, reanalyzed existing experimental data from a trace conditioning task in Coddington and Dudman (2018)[32]. The horizontal axis is the training trial, and the vertical axis is the mean activity modulation of DA neuron activity integrated over both the cue and reward periods (relative to baseline). Each blue dot represents a recording period for an individual neuron from either the ventral tegmental area (VTA) or substantia nigra compacta (SNc) ($n = 96$). The black line is a running average over 10 trials. A bracket with a star indicates blocks of 10 individual cell recording periods (dots) which show a significantly different modulated DA response (integrated over both the cue and reward periods) than that of the first 10 recording periods/cells (Significance with a two-sided Wilcoxon rank sum test, $p < 0.05$). See also Supplementary Fig. 1. Definitions: temporal difference learning (TD), dopamine neurons (DA).

moving bump is observed[30]. Taken together, these different results suggest that at least in some cases a moving bump is not observed. However, since a moving bump is not predicted in TD($\lambda$) for sufficiently large $\lambda$, these results do not invalidate the TD framework in general, but rather suggest that in some cases at least the TD(0) variant is inconsistent with the data[28].

While the moving bump prediction is parameter dependent, another prediction common to all TD variants is that the integrated RPE, obtained by summing response magnitudes over the whole trial duration, does not exceed the initial US response on the first trial. This prediction is robust because the normative basis of TD is to evaluate expected reward or discounted expected reward. In versions of TD where non-discounted reward is evaluated ($\gamma = 1$) the integral of RPE activity should remain constant throughout learning (see Supplementary Material Appendix A for proof, Supplementary Fig. 1 for simulations). Commonly TD estimates discounted rewards ($\gamma < 1$),

where the discount means that rewards that come with a small delay are worth more than rewards that arrive with a large delay. With discounted rewards, the integral of RPE activity will decrease with learning and become smaller than the initial US response. In contrast, we reanalyzed data from several recent experiments[29,31,32] and found that the integrated response can transiently increase over the course of learning (Fig. 3d, Supplementary Fig. 1).

An additional prediction of TD learning which holds across many variants is that when learning converges, and if a reward is always delivered (100% reward delivery schedule), the response of RPE neurons at the time of reward is zero. Even in the case where a small number of rewards are omitted (e.g., 10%), TD predicts that the response of RPE neurons at the time of reward is very small, much smaller than at the time of the stimulus. This seems to be indeed the case for several example neurons shown in the original experiments of dopaminergic VTA neurons[5]. However, additional data obtained more recently

indicates this might not always be the case and that significant response to reward, at the time of reward, persists throughout training[28,29,33]. This discrepancy between TD and the neural data is observed both for experiments in which responses throughout learning are presented[28,29] as well as in experiments that only show results after training[33].

In experimental approaches affording large ensembles of DA neurons to be simultaneously recorded, a diversity of responses has been reported. Some DA neurons are observed to become fully unresponsive at the time of reward, while others exhibit a robust response at the time of reward that is no weaker than the initial US response of these cells. This is clearly exhibited by one class of dopaminergic cells (type I) that Cohen et. al.[33] recorded in VTA. This diversity implies that TD is inconsistent with the results of some of the recorded neurons, but it is possible that TD does apply to the whole population (see work regarding "distributional TD"[34,35]). One complicating factor is that in most experiments we have no way of ascertaining that learning has reached its final state.

The original conception of TD is clean, elegant, and based on a simple normative foundation of estimating discounted expected rewards. Over the years, various experimental results that do not fully conform with the predictions of TD have been interpreted as consistent with the theory by making significant modifications to the classical value-based formulation of TD[34,36]. Such modifications might include an assumption of different variants of the learning rule for each neuron, such that each dopaminergic neuron no longer represents a formal, value-based RPE[34], or an assumption of additional inputs such that even when the expectation of reward is learned and fully expected dopaminergic neurons still respond at the time of reward. Note that we are not suggesting that such modifications are incorrect or biophysically implausible – on the contrary, we are suggesting that the success of these models demonstrates that the original, normative marriage between TD, RPE, and DA should be open to scrutiny.

Towards this end, a recent paper has also shown that dopamine release, as recorded with photometry, seems to be inconsistent with RPE[30]. This paper has shown many experimental results that are at odds with those expected by an RPE, and specifically, these experiments show that dopamine release at least partially represents the retrospective probability of stimulus-given reward. Other work has suggested that dopamine signaling is more consistent with direct learning of behavioral policy than a value-based RPE[37]. The culmination of these results leads us to consider new theories that may underlie the mechanisms behind observed DA release in the brain.

## The FLEX theory of reinforcement learning: A theoretical framework based on a plastic temporal basis

We propose an alternative theoretical framework underlying observed dopamine dynamics, coined FLEX (**F**lexibly **L**earned **E**rrors in E**x**pected Reward). The FLEX theory assumes that there is a plastic (as opposed to fixed) temporal basis that evolves alongside the changing response of reward-dependent neurons (such as DA neurons in VTA). The theory in general is agnostic about the functional form of the temporal basis, and several possible examples are shown in Fig. 4 (top, schematic; middle, characteristic single unit activity; bottom, population activity before and after learning). Feed-forward neural sequences[38,39] (Fig. 4a), homogenous recurrent networks (Fig. 4b), and heterogeneous recurrent networks (Fig. 4c) could all plausibly support the temporal basis. Before learning, well-developed basis-functions do not exist, though some neurons do respond transiently to the CS. Over learning, basis-functions develop (Fig. 4, bottom) in response to the rewarded stimulus, but not to unrewarded stimuli. In FLEX, we discard the implausible assumption of a separate, predeveloped basis for every possible stimulus that spans an arbitrary amount of time. Instead, basis functions only form in the process of learning, develop only to stimuli that are tied to reward, and only span the relevant temporal interval for the behavior.

In the following, we demonstrate that such a framework can be implemented in a biophysically plausible model and that such a model not only agrees with many existing experimental observations but also can reconcile seemingly incongruent results pertaining to sequential conditioning. The aim of this model is to show that the FLEX theoretical framework is possible and plausible given the available data, not to claim that this implementation is a perfectly validated model of reinforcement learning in the brain. Previous models concerning hippocampus and prefrontal cortex (PFC) have also considered cue memories with adaptive durations, but not explicitly in the context of challenging the fundamental idea of a fixed temporal basis[40,41].

## A biophysically plausible implementation of FLEX, proof-of-concept

Here we present a biophysically plausible proof-of-concept model that implements FLEX. This model is motivated by previous experimental results[23,24] and previous theoretical work[42–46]. The network's full architecture (visualized in Fig. 5a) consists of two separate modules, a basis function module, and a reward module, here mapped onto distinct brain areas. We treat the reward module as an analogue of the VTA and the basis-function module akin to a cortical region such as the mPFC or OFC (although other cortical or subcortical regions, notably striatum[22,47] might support temporal basis functions). All cells in these regions are modeled as spiking integrate-and-fire neurons (see Methods).

We assume that within our basis function module are subpopulations of neurons tuned to certain external inputs, visualized in Fig. 5a as a set of discrete "columns", each responding to a specific stimulus. Within each column there are both excitatory and inhibitory cells, with a connectivity structure that includes both plastic (dashed lines, Fig. 5a) and fixed synaptic connections (solid lines, Fig. 5a). The VTA is composed of dopaminergic (DA) and inhibitory GABAergic cells. The VTA neurons have a background firing rate of ~5 Hz, and the DA neurons have preexisting inputs from "intrinsically rewarding" stimuli (such as a water reward). The plastic and fixed connections between the modules and from both the CS and US to these modules are also depicted in Fig. 5a.

The model's structure is motivated by observations of distinct classes of temporally-sensitive cell responses that evolve during trace conditioning experiments in medial prefrontal cortex (mPFC), orbitofrontal cortex (OFC) and primary visual cortex (V1)[23,24,45,48,49]. The architecture described above allows us to incorporate these observed cell classes into our basis-function module (Fig. 5b). The first class of neurons ("Timers") are feature-specific and learn to maintain persistently elevated activity that spans the delay period between cue and reward, eventually decaying at the time of reward (real or expected). The second class, the "Messengers", has an activity profile that peaks at the time of real or expected reward. This set of cells forms what has been coined a "Core Neural Architecture" (CNA)[44], a potentially canonical package of neural temporal representations. A slew of previous studies have shown these cell classes within the CNA to be a robust phenomenon experimentally[24,45,48–52], and computational work has demonstrated that the CNA can be used to learn and recall single temporal intervals[44], Markovian and non-Markovian sequences[43,53]. For simplicity, our model treats connections between populations within a single CNA as fixed (previous work has shown that such a construction is robust to the perturbation of these weights[43,44]).

Learning in the model is dictated by the interaction of eligibility traces and dopaminergic reinforcement. We use a previously established two-trace learning rule[43,45,46,54] (TTL), which assumes two Hebbian-activated eligibility traces, one associated with LTP and one associated with LTD (see Methods). We use this rule because it solves the temporal credit assignment problem inherent in trace conditioning, reaches stable fixed points, and since such traces

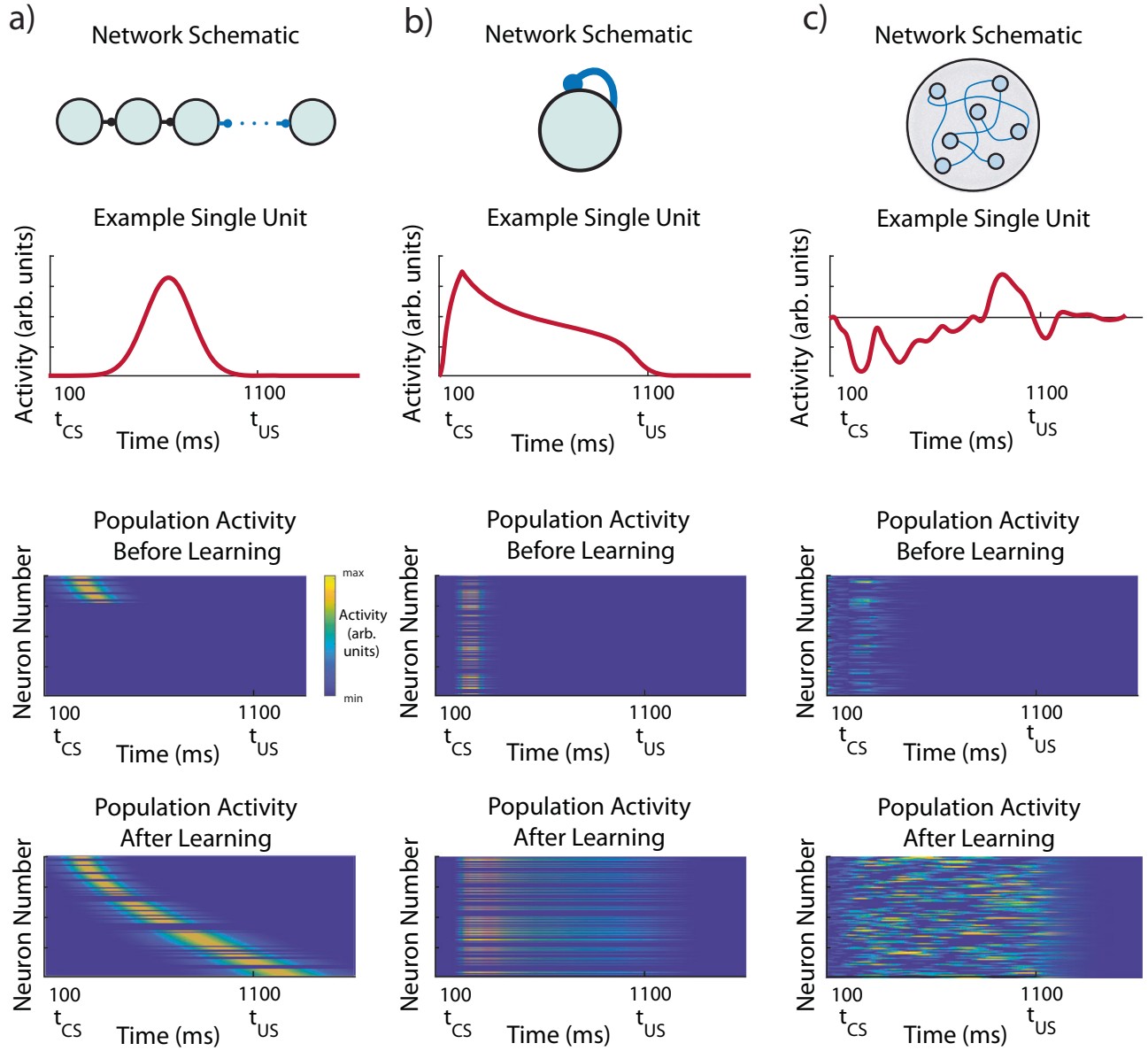

**Fig. 4 | Potential architectures for a flexible temporal basis.** Three example networks that could implement a FLEX theory. Top, network schematic. Middle, the activity of one example neuron in the network. Bottom, network activity before and after training. Each network initially only has transient responses to stimuli, modifying plastic connections (in blue) during association to develop a specific temporal basis to reward-predictive stimuli. **a** Feed-orward neural sequences or "chains" could support a FLEX model, if the chain could recruit more members during learning,

exclusive to reward-predictive stimuli. **b** A population of neurons with homogenous recurrent connections have a characteristic decay time that is related to the strength of the weights. The cue-relative time can then be read out by the mean level of activity in the network. **c** A population of neurons with heterogenous and large recurrent connections (liquid state machine) can represent cue-relative time by treating the activity vector at time t as the "microstate" representing time t (as opposed to the homogenous case, where only mean activity is used).

have been experimentally observed in trace conditioning tasks[45]. In theory, other methods capable of solving the temporal credit assignment problem (such as a rule with a single eligibility trace[42]) could also be used to facilitate learning in FLEX, but owing to its functionality and experimental support, we choose to utilize TTL for this work. See the Methods section for details of the implementation of TTL used here.

Now, we will use this implementation of FLEX to simulate several experimental paradigms, showing that it can account for reported results. Importantly, some of the predictions of the model are categorically different than those produced by TD, which allows us to distinguish between the two theories based on experimental evidence.

### CS-evoked and US-evoked Dopamine responses evolve on different timescales

First, we test FLEX on a basic trace conditioning task, where a single conditioned stimulus is presented, followed by an unconditioned stimulus at a fixed delay of one second (Fig. 6a). The evolution of FLEX over training is mediated by reinforcement learning (via TTL) in three sets of weights: Timer → Timer, Messenger → VTA GABA neurons, and CS→VTA DA neurons. These learned connections encode the feature-specific cue-reward delay, the temporally specific suppression of US-evoked dopamine, and the emergence of CS-evoked dopamine, respectively.

Upon presentation of the cue, cue neurons (CS) and feature-specific Timers are excited, producing Hebbian-activated eligibility

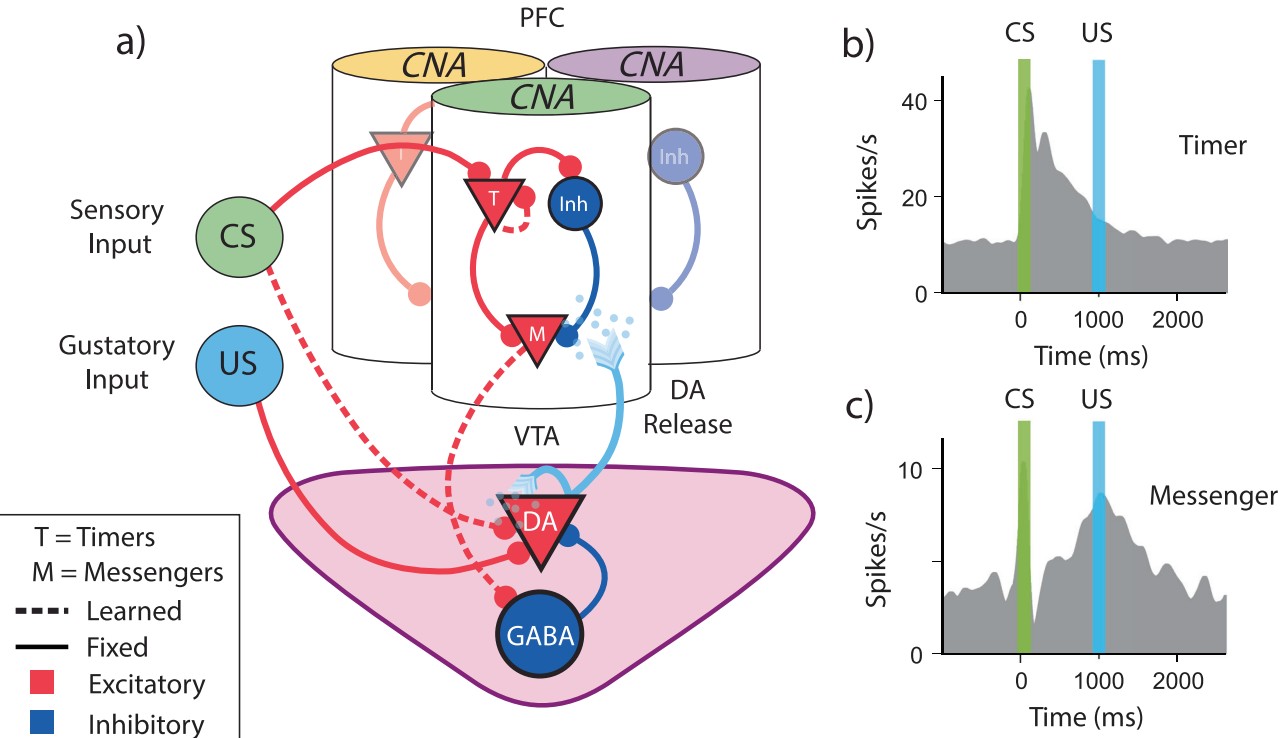

**Fig. 5 | Biophysically Inspired Architecture Allows for Flexible Encoding of Time. a** Diagram of the model architecture. Core neural architectures (CNAs, visualized here as columns) located in the PFC are selective to certain sensory stimuli (indicated here by color atop the column) via fixed excitatory inputs (conditioned stimulus, CS). Ventral tegmental area dopamine (VTA, DA) neurons receive fixed input from naturally positive valence stimuli, such as food or water reward (unconditioned stimulus, US). DA neuron firing releases dopamine, which acts as a learning signal for both PFC and VTA. Solid lines indicate fixed connections, while dotted lines indicate learned connections. **b**, **c** Schematic representation of data adapted from Liu et al. [51]. **b** Timers learn to characteristically decay at the time of cue-predicted reward. **c** Messengers learn to have a firing peak at the time of cue-predicted reward. Definitions: prefrontal cortex (PFC), ventral tegmental area (VTA).

traces at their CS→DA and T→T synapses, respectively. When the reward is subsequently presented one second later, the excess dopamine it triggers acts as a reinforcement signal for these eligibility traces (which we model as a function of the DA neuron firing rate, D(t), see Methods) causing both the cue neurons' feed-forward connections and the Timers' recurrent connections to increase (Fig. 6b).

Over repeated trials of this cue-reward pairing, the Timers' recurrent connections continue to increase until they approach their fixed points, which corresponds to the Timers' persistent firing duration increasing until it spans the delay between CS and US (Supplementary Fig. 2 and Methods). These mature Timers then provide a feature-specific representation of the expected cue-reward delay.

The increase of feed-forward connections from the CS to the DA neurons (Fig. 6c) causes the model to develop a CS-evoked dopamine response. Again, this feed-forward learning uses dopamine release at $t_{US}$ as the reinforcement signal to convert the Hebbian activated CS→DA eligibility traces into synaptic changes (Supplementary Fig. 3). The emergence of excess dopamine at the time of the CS ($t_{CS}$) owing to these potentiated connections also acts to maintain them at a non-zero fixed point, so CS-evoked dopamine persists long after US-evoked dopamine has been suppressed to baseline (see Methods).

As the Timer population modifies its timing to span the delay period, the Messengers are "dragged along", since, owing to the dynamics of the Messengers' inputs (T and Inh), the Messengers themselves selectively fire at the end of the Timers' firing envelope. Eventually, the Messengers overlap with the tonic background activity of VTA GABAergic neurons at the time of the US ($t_{US}$) (Fig. 6b). When combined with the dopamine release at $t_{US}$, this overlap triggers Hebbian learning at the Messenger → VTA GABA synapses (see Fig. 6c, Methods), which indirectly suppresses the DA neurons. Because of the

temporal specificity of the Messengers, this learned inhibition of the DA neurons (through excitation of the VTA GABAergic neurons) is effectively restricted to a short time window around the US and acts to suppress DA neural activity at $t_{US}$ back towards baseline.

As a result of these processes, our model recaptures the traditional picture of DA neuron activity before and after learning a trace conditioning task (Fig. 6b). While the classical single neuron results of Schultz and others suggested that DA neurons are almost completely lacking excess firing at the time of expected reward[5], more recent calcium imaging studies have revealed that a complete suppression of the US response is not universal. Rather, many optogenetically identified dopamine neurons maintain a response to the US and show varying development of a response to the CS[28,29,33]. This diversity is also exhibited in our implementation of FLEX (Fig. 6d) due to the connectivity structure which is based on sparse random projections from the CS to the VTA and from the US to VTA.

During trace conditioning in FLEX, the inhibition of the US-evoked dopamine response (via M→GABA learning) occurs only after the Timers have learned the delay period (since M and GABA firing must overlap to trigger learning), giving the potentiation of the CS response time to occur first. At intermediate learning stages (e.g. trial 5, Fig. 6a, b), the CS-evoked dopamine response (or equivalently, the CS→DA weights) already exhibits significant potentiation while the US-evoked dopamine response (or equivalently, inverse of the M → US weights) has only been slightly depressed. While this phenomenon has been occasionally observed in certain experimental paradigms[28,32,55,56], it has not been widely commented on – in FLEX, this is a fundamental property of the dynamics of learning (in particular, very early learning).

If an expected reward is omitted in FLEX, the resulting DA neuron firing will be inhibited at that time, demonstrating the

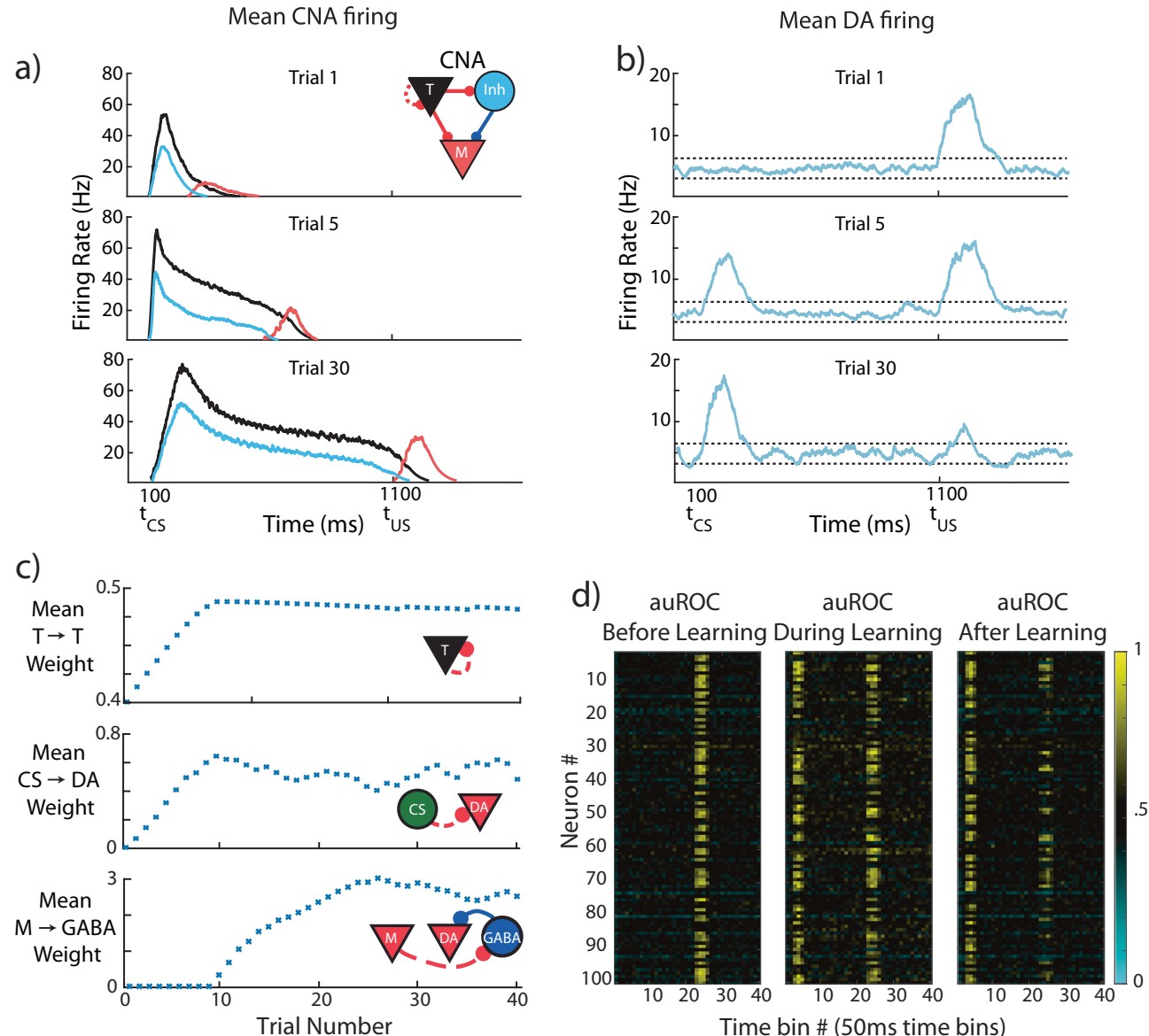

**Fig. 6 | CS-evoked and US-evoked model dopamine responses evolve on different timescales.** The model is trained for 30 trials while being presented with a conditioned stimulus (CS) at 100 ms and a reward at 1100 ms. **a** Mean firing rates for the core neural architecture (CNA) (see inset for colors; T = Timers, M = Messengers, Inh = Inhibitory), for three different stages of learning. **b** Mean firing rate over all DA neurons taken at the same three stages of learning. Firing above or below thresholds (dotted lines) evokes positive or negative D(t) in PFC. **c** Evolution of mean synaptic weights over the course of learning. Top, middle, and bottom, mean strength of Timer→Timer, CS → DA (conditioned stimulus → dopamine), and Messenger→GABA synapses, respectively. **d** Area under receiver operating characteristic (auROC, see Methods) for all VTA neurons in our model for 15 trials before (left, unconditioned stimulus, US, only), 15 trials during (middle, CS + US, conditioned + unconditioned stimulus), and 15 trials after (right, CS + US) learning. Values above and below 0.5 indicate firing rates above and below the baseline distribution. Definitions: dopamine (DA), prefrontal cortex (PFC), inhibitory neurons (GABA).

characteristic dopamine "dip" seen in experiments (Supplementary Fig. 4)[5]. This phenomenon occurs in our model because the previous balance between excitation and inhibition (from the US and GABA neurons, respectively) is disrupted when the US is not presented. The remaining input at $t_{US}$ is therefore largely inhibitory, resulting in a transient drop in firing rates. If the CS is consistently presented without being paired with the US, the association between cue and reward is unlearned, since the consistent negative D(t) at the time of the US causes depression of CS → DA weights (Supplementary Fig. 4). Additionally, unlike fixed RNN models, FLEX is able to learn and report accurate stimulus reward associations even in the presence of distractor cues, both during and after training (Supplementary Fig. 5). Further, although the implementation of FLEX we discuss here in the main text does not explicitly report value, with

slight modifications to our assumptions of the persistence of DA or other neuromodulators (see Methods), FLEX can account for the scaling of cue magnitude with initial reward value (Supplementary Fig. 6).

## Dynamics of FLEX diverge from those of TD during conditioning

FLEX's property that the evolution of the CS responses can occur independently of (and before) depression of the US response underlies a much more fundamental and general departure of our model from TD-based models. In our model, DA activity does not "travel" backwards over trials[1,5] as in TD(0), nor is DA activity transferred from one time to the other in an equal and opposite manner as in TD(λ)[11,28]. This is because our DA activity is not a strict RPE signal. Instead, while the DA neural firing in FLEX may resemble RPE following successful

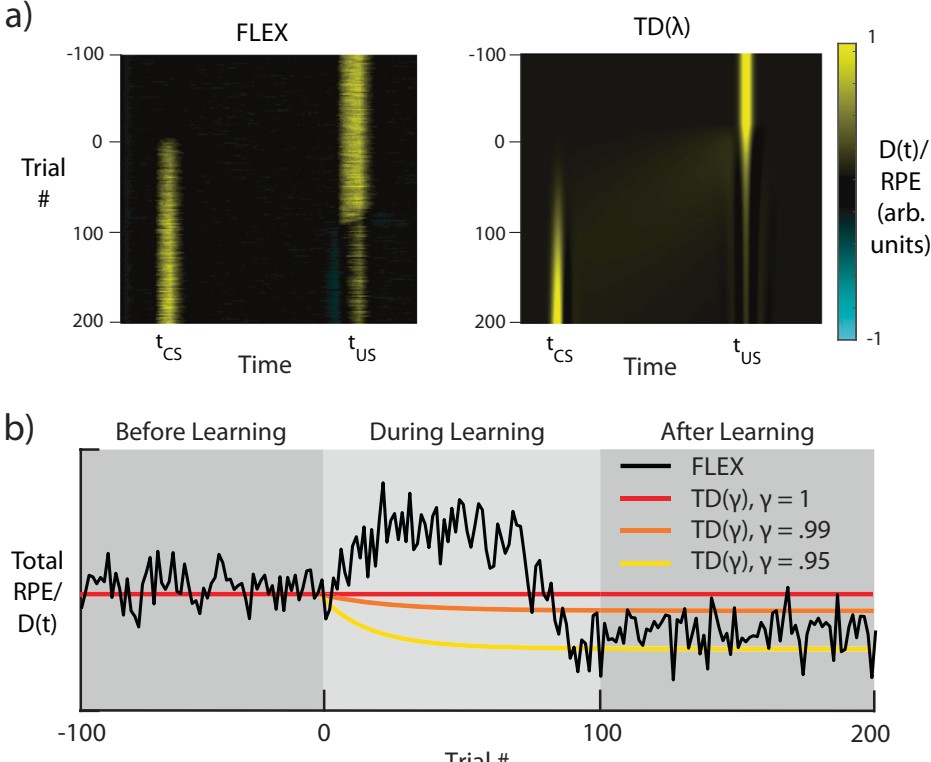

**Fig. 7 | FLEX Model Dynamics Diverge from those of TD Learning.** Dynamics of both temporal difference (TD) learning with $\lambda = 0.975$ (TD($\lambda$)) and FLEX when trained with the same conditioning protocol as shown in Fig. 6. **a** Dopaminergic release D(t) for FLEX (left), and RPE for TD($\lambda$) (right), over the course of training. **b** Total/integrated reinforcement D(t) during a given trial of training in FLEX (black), and sum total RPE in three instances of TD($\lambda$) with discounting parameters $\gamma = 1$, $\gamma = .99$, and $\gamma = .95$ (red, orange, and yellow, respectively). Shaded areas indicate functionally different stages of learning in FLEX. The learning rate in our model is reduced in this example, to make a direct comparison with TD($\lambda$). 100 trials of US-only presentation (before learning) are included for comparison with subsequent stages.

learning, the DA neural firing and RPE are not equivalent, as evidenced during the learning period.

To demonstrate this, we compare the putative DA responses in FLEX to the RPEs in TD($\lambda$) (Fig. 7a), training on the previously described trace conditioning task (see Fig. 6a). We set the parameters of our TD($\lambda$) model to match those in earlier work[28] and approximate the cue and reward as Gaussians centered at $t_{CS1}$ and $t_{CS2}$, respectively. In TD models, by definition, the integral of the RPE over the course of the trial is always less than or equal to the original total RPE provided by the initial unexpected presentation of reward. In other words, the error in reward expectation cannot be larger than the initial reward. In both TD(0) and TD($\lambda$), this quantity of "integrated RPE" is conserved; for versions of TD with a temporal discounting factor $\gamma$ (which acts such that a reward with value $r_0$ presented n timesteps in the future is only worth $r_0\gamma^n$ where $\gamma \le 1$), this quantity decreases as learning progresses (see Supplementary Material Appendix A and for proof and Supplementary Fig. 1 for simulations).

In FLEX, by contrast, integrated dopaminergic release D(t) during a given trial can be greater than that evoked by the original unexpected US (see Fig. 7b), and therefore during training the DA signal in FLEX diverges from a reward prediction error. This property has not been explicitly investigated, and most published experiments do not provide continuous data during the training phase. However, to test our prediction, we re-analyzed recently published data which does cover the training phase[29,32], and found that there is indeed a significant transient increase in dopamine release during training (Fig. 3d and Supplementary Fig. 1). Another recent publication found that initial DA response to the US was uncorrelated with the final DA response to the

CS[37], which also supports the idea that integrated dopamine release is not conserved.

## FLEX unifies sequential conditioning results
Standard trace conditioning experiments with multiple cues (CS1→CS2→US) have generally reported the so-called "serial transfer of activation" – that dopamine neurons learn to fire at the time of the earliest reward-predictive cue, "transferring" their initial activation at $t_{US}$ back to the time of the first conditioned stimulus[57,58]. However, other results have shown that the DA neural responses at the times of the first and second conditioned stimuli ($t_{CS1}$ and $t_{CS2}$, respectively) evolve together, with both CS1 and CS2 predictive of the reward[28,30].

Surprisingly, FLEX can reconcile these seemingly contradictory results. In Fig. 8 we show simulations of sequential conditioning using FLEX. In early training, we observe an emerging response to both CS1 and CS2, as well as to the US (Fig. 8b, ii). Later on, the response to the US is suppressed (Fig. 8b, iii). During late training (Fig. 8b, iv) the response to CS2 is suppressed and all activation is transferred to the earliest predictive stimulus, CS1. These evolving dynamics can be compared to the different experimental results. Early training is similar to the results of Pan et al. [28] and Jeong et al. [30], as seen in Fig. 8c, while the late training results are similar to the results of Schultz (1993)[57] as seen in Fig. 8d.

After training, in FLEX, both sequential cues still affect the dopamine release at the time of reward, as removal of either cue results in a partial recovery of the dopamine response at $t_{US}$ (Supplementary Fig. 7). This is the case even in late training in FLEX, when there is a positive dopamine response only to first cue – our model predicts that removal of the second cue will result in a positive dopamine response

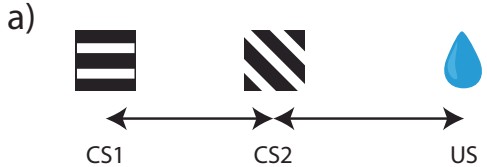

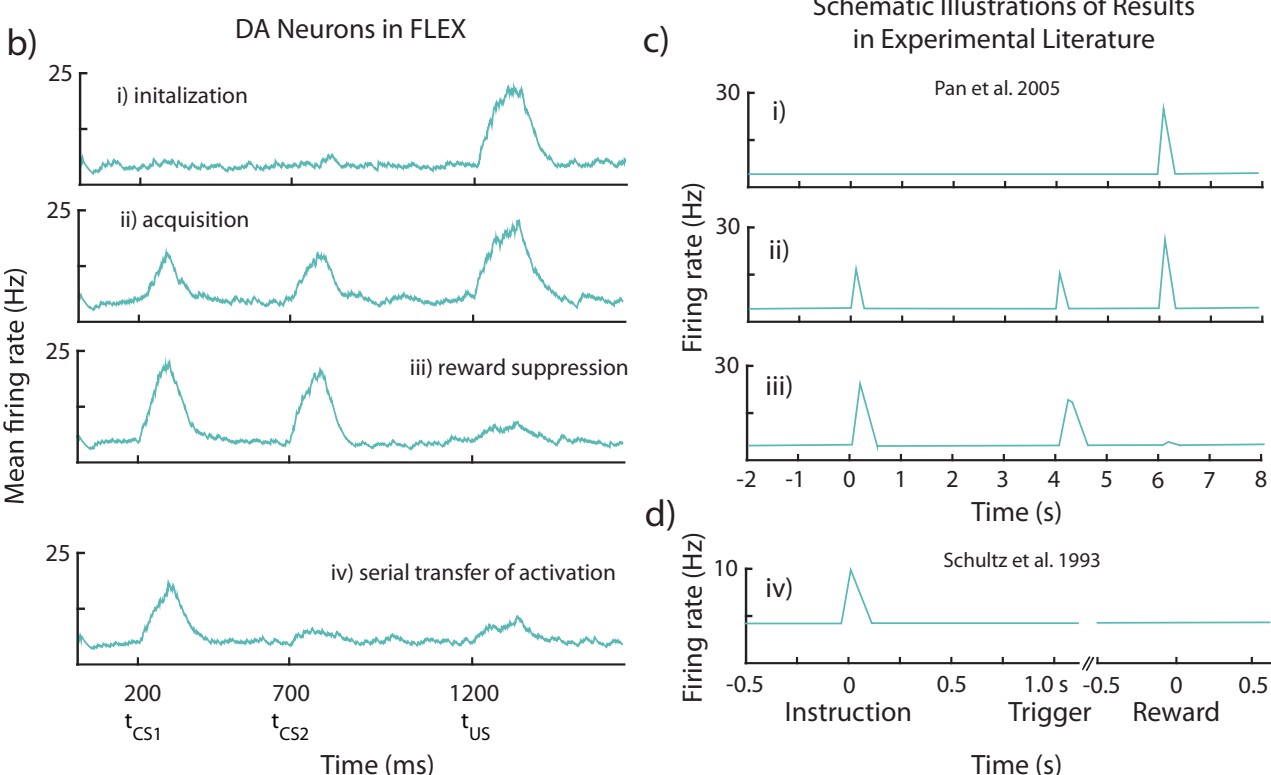

**Fig. 8 | FLEX model reconciles differing experimental phenomena observed during sequential conditioning.** Results from "sequential conditioning", where sequential neutral stimuli CS1 and CS2 (conditioned stimuli 1 and 2) are paired with delayed reward (US). **a** Visualization of the protocol. In this example, the US is presented starting at 1500 ms, with CS1 is presented starting at 100 ms, and CS2 is presented starting at 800 ms. **b** Mean firing rates over all dopamine (DA) neurons, for four distinctive stages in learning – initialization(i), acquisition(ii), reward depression(iii), and serial transfer of activation(iv). **c** Schematic illustration of experimental results from recorded dopamine neurons, labeled with the matching stage of learning in our model. **c** DA neuron firing before (top), during (middle), and after (bottom) training, wherein two cues (0 s and 4 s) were followed by a single reward (6 s). Adapted from Pan et al. [28]. **d** DA neuron firing after training wherein two cues (instruction, trigger) were followed by a single reward. Adapted from Schultz et al. [57].

at the time of expected reward. In contrast, the RPE hypothesis would posit that after extended training, the value function would eventually be maximized following the first cue, and therefore removal of the subsequent cue would not change dopamine release at $t_{US}$.

FLEX is also capable of replicating the results of a different set of sequential learning paradigms (Supplementary Fig. 8). In these protocols, the network is initially trained on a standard trace conditioning task with a single CS. Once the cue-reward association is learned completely, a second cue is inserted in between the initial cue and the reward, and learning is repeated. As in experiments, this intermediate cue on its own does not become reward predictive, a phenomenon called "blocking"[59–61]. However, if reward magnitude is increased or additional dopamine is introduced to the system, a response to the intermediate CS (CS2) emerges, a phenomenon termed "unblocking"[62,63]. Each of these phenomena can be replicated in FLEX (Supplementary Fig. 8).

## Discussion

TD has established itself as one of the most successful models of neural function to date, as its predictions regarding RPE have, to a large extent, matched experimental results. However, two key factors make it reasonable to consider alternatives to TD as a modeling framework for how midbrain dopamine neurons could learn RPE. First, attempts for biologically plausible implementations of TD have previously assumed that even before learning, each possible cue triggers a separate chain of neurons which tile an arbitrary period of time relative to the cue start. The a priori existence of such an immense set of fixed, arbitrarily long cue-dependent basis functions is both fundamentally implausible and inconsistent with experimental evidence which demonstrates the development or maturation of temporal bases over learning[22–25,45,48,49]. Second, different conditioning paradigms have revealed dopamine dynamics that are incompatible with the predictions of models based on the TD framework[28,30,64,65].

To overcome these problems, we suggest that the temporal basis itself is not fixed, but instead plastic, and is learned only for specific cues that lead to reward. We call this theoretical framework FLEX. We also presented a biophysically plausible implementation of FLEX, and we have shown that it can generate flexible basis functions and that it produces dopamine cell dynamics that are consistent with experimental results. Our implementation should be seen as a proof-of-concept model. It shows that FLEX can be implemented with biophysical components and that such an implementation is consistent with much of the data. It does not show that the specific details of this

implementation, (including the brain regions in which the temporal-basis is developed, the specific dynamics of the temporal basis functions, and the learning rules used) are those used by the brain.

Cue reward associations can be behaviorally learned even for long durations between the cue and reward. The specific implementation of FLEX proposed here can only learn them for a span of up to several seconds. This limitation stems from two aspects of the model. First, the spiking stochastic recurrent network itself cannot reliably express temporal intervals of durations that are longer than a few seconds. Second, the synaptic eligibility traces only last several seconds[45,66]. We have analyzed these limitations in a previous paper and showed that by including additional active intrinsic conductance's in the neuronal model these limitations can be relaxed and time gaps of tens of seconds can be learned[67]. For simplicity, we have not included these additional mechanisms here. For longer time gaps on the order of minutes, different types of mechanisms must be considered, though they too might fall into the general FLEX framework.

One of the appealing aspects of TD learning is that it arises from the simple normative assumption that the brain needs to estimate future expected rewards. Does FLEX have a similar normative basis? Indeed, in the final stage of learning, the responses of DA neurons in FLEX resemble RPE neurons in TD, however, in FLEX there is no analog for the value neurons assumed in TD. Unlike in TD, the activity of FLEX DA neurons in response to the cue represents an association with future expected rewards, independent of a valuation. Though a slight modification of FLEX (see Methods and Supplementary Fig. 6) is able to learn value, the observation that FLEX does not require value neurons in order to learn associations or even the value of expected reward, does not mean such neurons do not exist in the brain, or are not useful. Although experiments have reported value neurons in the brain[68], it has been debated whether value neurons indeed exist and whether they are useful[69], but this is not part of the FLEX theory.

Instead of primarily learning value, the goal of FLEX is to learn the association between cue and reward, develop the temporal basis functions that span the period between the two, and transfer DA signaling from the time of the reward to the time of the cue. These basic functions could then be used as a mechanistic foundation for brain activities, including the timing of actions. Once these basis functions exist it might be possible to use them for future TD type learning. In essence, DA in FLEX acts to create internal models of associations and timings. Recent experimental evidence[37] which suggests that DA correlates more with the direct learning of behavioral policies rather than value-encoded prediction errors is broadly consistent with this view of dopamine's function within the FLEX framework.

As noted, FLEX is a general theory, and there could be alternative models implementing it. Indeed, a recent publication[70] has proposed a similar model to account for VTA dynamics. This model also assumes that temporal associations learned in the cortex can explain the formation of VTA responses in the absence of a preexisting temporal basis. This previous publication differs from our model in the details of the recurrent network, which cell types inhibit the VTA response to reward, the details of the learning rules which are not based on biophysical observations, and in that it is a rate-based model, not a spiking model. It has also been applied to a somewhat different set of experimental observations. The major difference between the two papers is not the detailed differences in the mechanisms, but rather that our work explains explicitly the conceptual limitations of traditional models of TD learning, and how they fail to account for some experimental results. It then offers the general FLEX theory as an alternative to traditional TD algorithms.

Another recent publication[30] has challenged the claim that DA neurons indeed represent RPE, instead hypothesizing that DA release is also dependent on retrospective probabilities[30,71]. The design of most historical experiments cannot distinguish between

these competing hypotheses. In this recent research project, a set of new experiments was designed specifically to test these competing hypotheses, and the results obtained are inconsistent with the common interpretation that DA neurons simply represent RPE[30]. More generally, while the idea that the brain does and should estimate economic value seems intuitive, it has been recently questioned[69]. This challenge to the prevailing normative view is motivated by behavioral experimental results which instead suggest that a heuristic process, which does not faithfully represent value, often guides decisions.

Although recent papers have questioned the common normative view of the response of DA neurons in the brain and their relation to value estimation[13,30,37,69], the goal of this work is different. Here we survey problems with the implementation of TD algorithms in neuronal machinery, and propose an alternative theoretical formulation, FLEX, along with a computational implementation of this theory. The fundamental difference from previous work is that FLEX postulates that the temporal basis-functions necessary for learning are themselves learned, and that neuromodulator activity in the brain is an instructive signal for learning these basis functions. Our computational implementation has predictions that are different than those of TD and are consistent with many experimental results. Further, we tested a unique prediction of FLEX (that integrated DA release across a trial can change over learning) by re-analyzing experimental data, showing that the data was consistent with FLEX but not the TD framework.

Additional qualitative predictions of the specific computational model will require further experimental testing. First, our implementation of FLEX would predict that the temporal basis must form before the response at the US is inhibited. In the case of our model, with its Timer and Messenger cell types, this would mean that the duration of the Timer cells would gradually increase until it bridges the temporal gap between the CS and the US. Further, our implementation predicts that the activity of the Messenger cells shifts forward in time until it overlaps with the activity of the DA cells, and only subsequent to this might the activity of the DA cells at the time of reward be inhibited. These specific predictions still need to be tested, but note that these are predictions of our specific computational implementation of FLEX, not of the general FLEX framework. Other networks that could plausibly implement the flexible timing central to the FLEX theory (Fig. 4a, c) may not be consistent with these implementation-specific predictions.

## Methods

All simulations were run via custom code in MATLAB 2023b (see code availability statement). A full table of parameter values used for the main simulations is provided in Table 1.

### Network dynamics

The membrane dynamics for each neuron $i$ are described by the following equations:

$$C\frac{dv_i}{dt} = g_L(E_L - v_i) + g_{E,i}(E_E - v_i) + g_{I,i}(E_I - v_i) + \sigma \quad (1)$$

$$\frac{ds_i}{dt} = -\frac{s_i}{\tau_s} + \rho(1 - s_i)\sum_k \delta(t - t_k^i) \quad (2)$$

The membrane potential and synaptic activation of neuron $i$ are notated $v_i$ and $s_i$. $g$ refers to the conductance, $C$ the capacitance, and $E$ to the reversal potentials indicated by the appropriate subscript, where leak, excitatory, and inhibitory are indicated by subscripts $L$, $E$, and $I$, respectively. $\sigma$ is a mean zero noise term. The neuron spikes upon it crossing its membrane threshold potential $v_{th}$, after which it enters a refractory period $t_{ref}$. The synaptic activation $s_i$ is updated by an amount $\rho(1 - s_i)$, where $\rho$ is the fractional change in synaptic

## Table 1 | Model Parameters

| Parameter | Value | Units | Description |
|---|---|---|---|
| $N_{pp}$ | 100 | – | Number of E/I neurons per population |
| $N_{pop}$ | 3 | – | Number of E/I populations (Timer, Messenger, VTA) |
| $N$ | 300 | – | Number of E/I neurons in network |
| $N_{total}$ | 600 | – | Total number of neurons (E + I) in network |
| $n_{trials}$ | 30,80 | – | Number of training trials |
| $dt$ | 1 | ms | Integration timestep |
| $t_{cue}$ | 200 | ms | Time of cue |
| $t_{reward}$ | 1100 | ms | Time of reward |
| $D$ | 10 | ms | Intrinsic transmission delay |
| $p_r$ | 30 | Hz | Rate of Poisson stimulus pulse |
| $\sigma_N, \sigma_{VTA}$ | $N(0, 1e\text{-}10)$, $.0375 + N(0, 2.5e\text{-}3)$ | – | Gaussian white noise, Timer/Messenger neurons, VTA neurons |
| $\rho$ | 1/7 | – | Fractional change of synaptic activation |
| $\tau_w$ | 40 | ms | Time window for firing rate integration |
| $\tau_e, \tau_{ea}$ | 10, 20 | dt (arb.) | Time constants of eligibility traces |
| $\tau_s^E, \tau_s^I, \tau_s^{VTAe}, \tau_s^{VTAi}$ | 80, 20, 20, 10 | ms | Time constant for synaptic activation for excitatory (EE and IE), inhibitory (EI) connections, as well as VTA excitatory (EE) and VTA inhibitory connections (EI) |
| $g_L$ | 10 | nS | Leak conductance |
| $C_m$ | 200 | pF | Membrane capacitance |
| $E_L$ | –60 | mV | Leak reversal potential |
| $E_E, E_L, E_I$ | –5, –60, –70 | mV | Excitatory, leak, and inhibitory reversal potentials |
| $v_{th}, v_{th}^I$ | –55, –50 | mV | Spiking threshold potential (excitatory, inhibitory) |
| $v_{rest}$ | –60 | mV | Resting potential |
| $v_{hold}$ | –61 | mV | Reset potential |
| $t_{ref}$ | 3 | ms | Absolute refractory period (parameter value in MATLAB code is 2, because of implementation quirk, it is effectively 3) |
| $\tau_p, \tau_d$ | 1800, 800 | ms | LTP/LTD eligibility trace time constant, recurrent connections |
| $T_p^{max}, T_d^{max}$ | 0.003, 0.0033 | – | Saturation level, LTP/LTD eligibility trace, recurrent connections |
| $\eta_p, \eta_d$ | 300, 135 | ms$^{-1}$ | Activation rate, LTP/LTD eligibility trace, recurrent connections |
| $\tau_p^{FF}, \tau_d^{FF}$ | 2000, 800 | ms | LTP/LTD eligibility trace time constant, feed-forward connections |
| $T_p^{max,FF}, T_d^{max,FF}$ | 0.0015, 0.004 | – | Saturation level, LTP/LTD eligibility trace, feed-forward connections |
| $\eta_p^{FF}, \eta_d^{FF}$ | 650, 40 | ms$^{-1}$ | Activation rate, LTP/LTD eligibility trace, feed-forward connections |
| $d_0, \Delta d$ | 5, 2 | Hz | Firing rate threshold to trigger DA release above or below expectation (lower bound is d0 - Δd, upper bound is d0 + Δd) |
| $W_{EE}^{MT}, W_{EI}^{MT}$ | 0.5, –20 | nS | Synaptic connection strength, Timer to Messenger excitatory to excitatory (EE) and inhibitory to excitatory (EI) connections |
| $W_{EI}^{VTA,VTA}$ | –1.5 | nS | Synaptic connection strength, VTA-VTA inhibitory to excitatory (EI) connections |
| $W_{IE}^{TT}, W_{IE}^{MM}$ | 0.3, 1 | nS | Synaptic connection strength, Timer-Timer and Messenger-Messenger excitatory to inhibitory (IE) connections |
| $\eta_{rec,l}, \eta_{ff,l}, \eta_{VTA,l}$ | 0.00015, $N(0, 1.5)$, $N(0, 1e\text{-}2)$ | ms$^{-1}$ | Learning rates for recurrent, feed-forward, and Messenger-VTA connections, respectively |

activity, at each time ($t_k^i$) the neuron spikes, and decays exponentially with time constant $\tau_s$ when there is no spike.

The conductance $g$ is the product of the incoming synaptic weights and their respective presynaptic neurons:

$$g_{\alpha,i} = \sum_j W_{ij}^\alpha s_j \tag{3}$$

$W_{ij}^\alpha$ are the connection strengths from neuron $j$ to neuron $i$, where the superscript $\alpha$ can either indicate $E$ (excitatory) or $I$ (inhibitory). A firing rate estimate for each neuron $r_i$ is calculated as an exponential filter of the spikes, with a time constant $\tau_r$.

$$\tau_r \frac{dr_i}{dt} = -r_i + \sum_k \delta(t - t_k^i) \tag{4}$$

### Fixed RNN

For the network in Fig. 2, the dynamics of the units $u_i$ in the RNN are described by the equation below:

$$\tau_{net} \frac{du_i}{dt} = -u_i + \sum_k W_{ik} \phi(u_k) \tag{5}$$

Where $u_i$ are the firing rates of the units in the RNN, each with a time constant $\tau_{net}$. $W_{ik}$ are the recurrent weights of the RNN, each of which is drawn from a normal distribution $N(0, \frac{g}{\sqrt{K}})$, where g is the "gain" of the network[72] and $\phi$ is a sigmoidal activation function. Each of the inputs given to the network (A, B, or C) is a unique, normally distributed projection of a 100 ms step function.

### Dopaminergic Two-Trace Learning (dTTL)

Rather than using temporal difference learning, FLEX uses a previously established learning rule based on competitive eligibility traces, known

as "two-trace learning" or TTL[43,45,46]. However, we replace the general reinforcement signal R(t) of previous implementations with a dopaminergic reinforcement D(t). We repeat here the description of D(t) from the main text:

$$D(t) = \begin{cases} r_{DA}(t) - (r_0 - \theta), & r_{DA}(t) \leq r_0 - \theta \\ 0, & r_0 - \theta < r_{DA}(t) < r_0 + \theta \\ r_{DA}(t) - (r_0 + \theta), & r_{DA}(t) \geq r_0 + \theta \end{cases} \quad (6)$$

Where $r_{DA}(t)$ is the firing rate of DA neurons, $r_0$ is a baseline firing rate, and $\theta$ is a threshold for reinforcement. Note that the dopaminergic reinforcement can be both positive and negative, even though actual DA neuron firing (and the subsequent release of dopamine neurotransmitter) can itself only be positive. The bipolar nature of D(t) implicitly assumes that background tonic levels of dopamine do not modify weights, and that changes in synaptic efficacies are a result of a departure (positive or negative) from this background level. The neutral region around $r_0$ provides robustness to small fluctuations in firing rates which are inherent in spiking networks.

The eligibility traces, which are synapse-specific, act as long-lasting markers of Hebbian activity. The two traces are separated into LTP- and LTD-associated varieties via distinct dynamics, which are described in the equations below.

$$\tau^p \frac{dT_{ij}^p}{dt} = -T_{ij}^p + \eta^p H_{ij}\left(T_{max}^p - T_{ij}^p\right) \quad (7)$$

$$\tau^d \frac{dT_{ij}^d}{dt} = -T_{ij}^d + \eta^d H_{ij}\left(T_{max}^d - T_{ij}^d\right) \quad (8)$$

Here, $T_{ij}^a$ (where $a \in (p,d)$) is the LTP ($p$ superscript) or LTD ($d$ superscript) eligibility trace located at the synapse between the j-th presynaptic cell and the i-th postsynaptic cell. The Hebbian activity, $H_{ij}$, is a simple multiplication $r_i \cdot r_j$ for application of this rule in VTA, where $r_j$ and $r_i$ are the time-averaged firing rates at the pre- and post-synaptic cells. Experimentally, the "Hebbian" terms ($H_{ij}$) which impact LTP and LTD trace generation are complex[45], but in VTA we approximate with the simple multiplication $r_i \cdot r_j$. For synapses in PFC, we make the alteration that $H_{ij} = \frac{r_i \cdot r_j}{1 + \alpha D(t)}$, acting to restrict PFC trace generation for large positive RPEs. The alteration of $H_{ij}$ by large positive RPEs is inspired by recent experimental work showing that large positive RPEs act as event boundaries and disrupt across-boundary (but not within-boundary) associations and timing[73]. Functionally, this altered $H_{ij}$ biases Timers in PFC towards encoding a single delay period (cue to cue or cue to reward) and disrupts their ability to encode across-boundary delays.

The LTP and LTD traces activate (via activation constant $\eta^a$), saturate (at a level $T_{max}^a$), and decay (with time constant $\tau^a$) at different rates. D(t) binds to these eligibility traces, converting them into changes in synaptic weights. This conversion into synaptic changes is "competitive", being determined by the difference in the product:

$$\frac{dW_{ij}}{dt} = \eta D(t)\left(T_{ij}^p(t) - T_{ij}^d(t)\right) \quad (9)$$

where $\eta$ is the learning rate.

The above eligibility trace learning rules, with the inclusion of dopamine, are referred to as dopaminergic "two-trace" learning or dTTL. This rule is pertinent not only because it can solve the temporal credit assignment problem, allowing the network to associate events distal in time, but also because dTTL is supported by recent experiments that have found eligibility traces for in multiple brain regions[45,66,74–76]. Notably, in such experiments, the eligibility traces in prefrontal cortex were found to convert into positive synaptic changes

via delayed application of dopamine[45], which is the main assumption behind dTTL.

For simplicity and to reduce computational time, in the simulations shown, M → GABA connections are learned via a simple dopamine-modulated Hebbian rule, $\frac{dW_{ij}}{dt} = \eta D(t) r_i r_j$. Since these connections are responsible for inhibiting the DA neurons at the time of reward, this learning rule imposes its own fixed point by suppressing D(t) down to 0. For any appropriate selection of feed-forward learning parameters in dTTL (Eq. 9), the fixed-point D(t) = 0 is reached well before the fixed point $T_{ij}^p(t) = T_{ij}^d(t)$. This is because, by construction, the function of M → GABA learning is to suppress D(t) down to zero. Therefore, the fixed point $T_{ij}^p(t) = T_{ij}^d(t)$ needs to be placed (via choosing trace parameters) beyond the fixed-point D(t) = 0. Functionally, then, both rules act to potentiate M → GABA connections monotonically until D(t) = 0. As a result, the dopamine-modulated Hebbian rule is in practice equivalent to dTTL in this case.

### Network Architecture
Our network architecture consists of two regions, VTA and PFC, each of which consists of subpopulations of leaky-integrate-and-fire (LIF) neurons. Both fixed and learned connections exist between certain populations to facilitate the functionality of our model.

To model VTA, we include 100 dopaminergic and 100 GABAergic neurons, both of which receive tonic noisy input to establish baseline firing rates of ~5 Hz. Naturally appetitive stimuli, such as food or water, are assumed to have fixed connections to DA neurons via the gustatory system. Dopamine release is determined by DA neurons firing above or below a threshold $\theta$, and dopamine release acts as reinforcement for all learned connections in the model.

Our model of PFC is comprised of different feature-specific 'columns'. Within each column there is a CNA microcircuit, with each subpopulation (Timers, Inhibitory, Messengers) consisting of 100 LIF neurons. Previous work has shown that these subpopulations can emerge from randomly distributed connections[44], and further that a single mean field neuron can well approximate the activity of each of these subpopulations of spiking neurons[77].

The two-trace learning rule we utilize for our model is described in further detail in previous work[43,45,46,54]. However, we will attempt to clarify how it functions below.

As a first approximation, the traces from the two-trace learning rule we utilize effectively act such that when they interact with dopamine above baseline, the learning rule will favor potentiation, and when they encounter dopamine below baseline, the learning rule will favor depression. Formally, the rule has fixed points which depend on both the dynamics of the traces and the dopamine release D(t):

$$\int_0^{T_{trial}} dt\, D(t)\left(T_{ij}^p(t) - T_{ij}^d(t)\right) = 0 \quad (10)$$

A trivial fixed point exists when D(t) = 0 for all t. Another simple fixed point exists in the limit that $D(t) = \delta(t_R - t)$, where $t_R$ is the time of reward, as Eq. 10 then reduces to $T_{ij}^p(t_R) = T_{ij}^d(t_R)$. In this case, the weights have reached their fixed point when the traces cross at the time of reward. In practice, the true fixed points of the model are a combination of these two factors (suppression of dopamine and crossing dynamics of the traces). In reality, D(t) is not a delta function (and may have multiple peaks during the trial), so to truly calculate the fixed points, one must use Eq. 10 as a whole. However, the delta approximation used above gives a functional intuition for the dynamics of learning in the model.

Supplementary Fig. 2 and Supplementary Fig. 3 demonstrate examples of fixed points for both recurrent and feed-forward learning, respectively. Note that in these examples the two "bumps" of excess dopamine (CS-evoked and US-evoked) are the only instances of non-zero

D(t). As such, we can take the integral in Eq. 10 and split it into two parts:

$$\int_{t_{CS,start}}^{t_{CS,and}} dt\, D(t)\left(T_{ij}^p(t) - T_{ij}^d(t)\right) + \int_{t_{US,start}}^{t_{US,end}} dt\, D(t)\left(T_{ij}^p(t) - T_{ij}^d(t)\right) = 0 \quad (11)$$

For recurrent learning, the dynamics evolve as follows. In the beginning, only the integral over $\Delta t_{US}$ exists, as D(t) is initially zero over $\Delta t_{CS}$ (Trial 1 in Supplementary Fig. 2). As a result, the learning rule evolves to approach the fixed point mediated by $\left(T_{ij}^p(t) - T_{ij}^d(t)\right)$ (Trial 20 in Supplementary Fig. 2). After the recurrent weights have reached this fixed point and the Timer neurons encode the cue-reward delay (Trial 30 in Supplementary Fig. 2), M → GABA learning acts to suppress D(t) down to zero as well (Trial 40 in Supplementary Fig. 2). Note again that we make the assumption that trace generation in PFC is inhibited during large positive RPEs. This acts to encourage the Timers to encode a single "duration" (whether cue-cue or cue-reward). In line with our assumption, experimental evidence has shown these large positive RPEs act as event boundaries and disrupt across-boundary (but not within-boundary) reports of timing[73].

For feed-forward learning, the weights initially evolve identically to the recurrent weights (Trial 1 in Supplementary Fig. 3). Again, only the integral over $\Delta t_{US}$ exists, so the feed-forward weights evolve according to $(T_{ij}^p(t) - T_{ij}^d(t))$. However, soon the potentiation of these feed-forward CS → DA weights themselves cause release of CS-evoked dopamine, and therefore we must consider both integrals to explain the learning dynamics (Trial 5 in Supplementary Fig. 3). This stage of learning is harder to intuit, but an intermediate fixed point is reached when the positive $\Delta W$ produced by the traces' overlap with US-evoked dopamine is equal and opposite to the negative $\Delta W$ produced by the traces' overlap with CS-evoked dopamine (Trial 20 in Supplementary Fig. 3). Finally, after US-evoked dopamine has been suppressed to baseline, the feed-forward weights reach a final fixed point where both positive and negative contributions to $\Delta W$ over the course of the CS offset each other (Trial 50 in Supplementary Fig. 3).

### Modified value-tracking network (Supplementary Fig. 6 only)

The above formulation of FLEX does not directly track the initial reward magnitude in the learned magnitude of the cue DA responses. This is due to the last part of feed forward learning (once the US-evoked DA has been suppressed to baseline), which leaves us with a fixed point which is independent of the initial reward magnitude. In order for the feed-forward weights (and thereby the cue-evoked DA response) to scale with the initial reward, we include an additional neuromodulatory term, A(t), which takes on the same functional form as our dopamine function D(t), but is not suppressed over the course of learning:

$$\frac{dW_{ij}}{dt} = \eta A(t)\left(T_{ij}^p(t) - T_{ij}^d(t)\right) \quad (12)$$

where η is the learning rate.

### Data Analysis

The measure of "area under receiver operating characteristic" (auROC) is used throughout this paper, for the purpose of making direct comparison to calcium imaging results that use auROC as a measure of statistical significance. Following the methods of Cohen et al.[33], time is tiled into 50 ms bins. For a single neuron, within each 50 ms bin, the distribution of spike counts for 25 trials of baseline spontaneous firing (no external stimuli) is compared to the distribution of spike counts during the same time bin for 25 trials of the learning phase in question. For example, in Fig. 6d, left, the baseline distributions of spike counts are compared to the distributions of spike counts when US only is presented. ROC is calculated for each bin by sliding the criteria from

zero to the max spike count within the bin, and then plotting P(active>criteria) versus P(baseline>criteria). The area under this curve is then a measure of discriminability between the two distributions, with an auROC of 1 demonstrating a maximally discriminable increase in spikes compared to baseline, 0 demonstrating a maximally discriminable decrease of spikes compared to baseline, and .5 demonstrating an inability to discriminate between the two distributions.

Data from Amo et al. was used for Supplementary Fig. 1[29,31]. The data from 7 animals (437-440, 444-446) is shown here. This dataset comes preprocessed, having been z-scored, "calculated from signals in an entire session smoothed with moving average of 50 ms"[28]. On every training day only the first 40 trials are used, unless there was a smaller number of rewarded trials, in which case that number was used. For every animal per every day, the integral from the time of the CS to the end of the trial is calculated and averaged over all trials in that day. These are the data points in Supplementary Fig. 1d, with each animal represented by a different color. The blue line is the average over animals. The distributions in Supplementary Fig. 1e represent the average over all animals in early (days 1–2), intermediate (day 3-4) and late (day 8-10) periods. Two-sided Wilcoxon rank sum tests find that intermediate is significantly higher than early ($p = 0.004$) and that late is significantly higher than early ($p = 0.006$). Late is not significantly lower than intermediate. We also find that late is significantly higher than early if we take the average per animal over the early and intermediate days ($p = 0.02$).

Supplementary Fig. 1f shows re-analyzed data from Coddington and Dudman (2018)[32]. The horizontal axis is the training trial, and the vertical axis is the mean activity modulation of DA neuron activity integrated over both the cue and reward periods (relative to baseline). Each blue dot represents a recording period for an individual neuron from either VTA or SNc ($n = 96$). The black line is a running average over 10 trials. A bracket with a star indicates blocks of 10 individual cell recording periods (dots) which show a significantly different modulated DA response (integrated over both the cue and reward periods) than that of the first 10 recording periods/cells(Significance with a two-sided Wilcoxon rank sum test, $p < 0.05$).

### Reporting summary

Further information on research design is available in the Nature Portfolio Reporting Summary linked to this article.

## Data availability

All data supporting the findings of this study are available within the paper and its accompanying custom MATLAB code (version 2023b). The code is available at https://github.com/ianconehed/FLEX[78]. Cone, I. Learning to Express Reward Prediction Error-like Dopaminergic Activity Requires Plastic Representations of Time. GitHub https://doi.org/10.5281/zenodo.11260815 (2024).

## Code availability

All simulations were run via custom code in MATLAB 2023b. The code is available at https://github.com/ianconehed/FLEX[78]. Cone, I. Learning to Express Reward Prediction Error-like Dopaminergic Activity Requires Plastic Representations of Time. GitHub https://doi.org/10.5281/zenodo.11260815 (2024).

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

## Acknowledgements

This work was supported by BBSRC BB/N013956/1 and BB/N019008/1 (C.C.), Wellcome Trust 200790/Z/16/Z (C.C.), Simons Foundation 564408 (C.C.), EPSRC EP/R035806/1 (C.C.), National Institute of Biomedical Imaging and Bioengineering 1R01EB022891-01 (H.S.), and Office of Naval Research N00014-16-R-BA01 (H.S.).

## Author contributions

I.C. and H.S. conceived the model. I.C., H.S. and C.C. designed the model. I.C. developed and performed the simulations. I.C., H.S. and C.C. wrote the manuscript.

## Competing interests

The authors declare no competing interests.
