## [Peer Review File · Nature Communications]

Learning to Express Reward Prediction Error-like Dopaminergic Activity Requires Plastic Representations of TimeReviewer #1 (Remarks to the Author):

In this manuscript, Cone et al. propose a model of error-based learning in which the temporal basis is not fixed during the course of learning but is itself learned. They show that the resultant algorithm produces a signal related to mesolimbic DA responses that deviate from some RPE predictions. In this sense, they contrast their model with TD RPE. They show that a biophysically plausible implementation of their model can capture several experimental results related to dopamine. Overall, I find that this work is valuable for the field, but has some limitations that dampen my enthusiasm. These are listed below.

1. Their core point here is that the temporal basis for associative learning need not be fixed. They show that a fixed basis can have some challenges. However, Hennig et al. showed recently that an RNN based model ("value RNN") can learn cue-reward associations without being given an explicit state space. Indeed, they show that the resultant latent representations appear like commonly assumed state spaces after learning. However, these representations are themselves learned. Thus, this model demonstrates that TD can work without a fixed temporal basis. How do the authors compare their model to this alternative?
2. A core point that the authors make is that when distractor stimuli are presented during a cue-reward simulation, an RNN will not develop a fixed basis set. As above, Hennig et al's RNN model does not have a fixed basis set but still learns the task. So, I am not sure that this is a convincing argument. Further, it is not clear to me that FLEX will naturally solve this task. Can the authors show that their model will learn the task shown in Figure 2 and naturally identify the distractors as being irrelevant?
3. How well does FLEX deal with long cue-reward delays such as those shown in experimental results (e.g., up to 16 s in some Schultz papers)?
4. While I am familiar with many similar claims of mismatch of RPE and experimental data, I am not familiar with the integral being less than 1. Can the authors mathematically demonstrate that TD-RPE must have an integral less than 1 for a Markov state space?
5. As best as I understand, FLEX is a model of learning associations and not value. A fundamental property of dopamine responses is scaling of cue responses by predicted reward magnitude. Can FLEX account for this?

Reviewer #2 (Remarks to the Author):

The prevailing hypothesis that dopamine (DA) represents temporal-difference (TD) reward prediction error (RPE) has been supported by many experimental results, but there are also a number of findings that appear inconsistent with the hypothesis, including that temporal shift of DA response from reward timing to cue timing (predicted from the hypothesis) is not always observed. Also, the DA = TD-RPE hypothesis presumes that each sensory cue stimulus evokes, by default, a specific temporal basis function, on which TD-RPE-based value learning occurs, but this presumption itself has a theoretical difficulty (it is practically impossible to generate basis function for "every" stimulus, but it is also impossible to generate it only for "relevant" stimulus without a priori knowledge about which stimulus is relevant). In order to address these difficulties in the traditional DA = TD-RPE hypothesis, the authors developed a new neurocomputational model, the FLEX model, that explains how such a stimulus-specific temporal basis function for reward-associated stimulus, initially not existing, can be gradually shaped through plasticity mechanisms. The authors showed that the model can be in line with experimental results that appear at odds with the DA = TD-RPE hypothesis.

I would agree with the authors that the issue tackled by this work is a central question in neuroscience or even widely in related sciences. I also like the FLEX model, its potential explanatory power and also its uniqueness as a model bridging the algorithmic level (TD reinforcement learning (RL)) and the implementation level (neuronal plasticity). However, I think there remain major points, described below, that are desired to be addressed or clarified.

- i) I expected that the FLEX model was first introduced at the algorithmic level (Marr's 2nd level), but the authors almost immediately went to the implementation level (Marr's 3rd level). After reading the whole article, I gradually understood that in the case of the FLEX model, these two

levels are really integrated together, and it may be difficult to describe its algorithm, as a set of equations, separately from implementation. But still, it would be helpful if the core mechanism of the model is first described before going to implementation. In the introduction, the authors explained when there are many sensory stimuli, A, B, C, etc, and only some of them, e.g., C, is associated with reward after a certain delay (unknown to agent) while other stimuli are just noise, the C-reward association is difficult to be learnt by fixed temporal basis function (or attractor dynamics) starting from each stimulus or general temporal basis function. These explanations are very nice, but what is lacking, I think, is a conceptual explanation of what the FLEX model is and how it can resolve this critical issue. In fact, there is already a corresponding description (section "The FLEX theory of reinforcement learning: A theoretical framework based on a plastic temporal basis.", page 10-11), but I think it is too broad and does not point to how the FLEX model resolves the critical issue (distinguishing C from other stimuli).

ii) Related to point i) above, I think it would be necessary to show the abovementioned point (i.e., activity starting from C is formed while activity starting from A or B is not) by actual simulation. Currently, the authors first described the case where there is only a single sensory stimulus (page 13-), and then described the case where there are sequentially presented multiple (high-order) cue stimuli (page 16-). But this latter case differs from the situation where there are many irrelevant stimuli (A, B, ..) and only some (C) is relevant, because all the sequentially presented stimuli are relevant (reward-associated) ones. I would like to see actual simulation results on how the FLEX model realizes formation of cue-specific sustained activation (temporal basis function) only for relevant stimuli.

iii) Regarding the point that cue-specific sustained activation (temporal basis function) is shaped through plasticity mechanisms, there exists a previous study that is not cited in the present manuscript:

Nicolas Deperrois, Victoria Moiseeva, & Boris Gutkin
Minimal Circuit Model of Reward Prediction Error Computations and Effects of Nicotinic Modulations
Front. Neural Circuits, Volume 12 - 2018
<https://doi.org/10.3389/fncir.2018.00116>

I think it would be necessary to cite this work and discuss how the FLEX model is conceptually different from (and/or superior to) this previous model (or if the authors think discussing this previous work is not necessary, please explain the reason). I think this point is especially important given that the authors claim that the FLEX model is a "proof-of-concept" model; how does it significantly depart from the concept of the previous work by Deperrois et al.?

iv) The authors' (FLEX model's) claim that DA does not represent TD-RPE by default but such a representation is developed through plasticity mechanisms is certainly an important departure from the traditional view. However, once DA's representation of TD-RPE is learnt, can it be used to implement TD RL (i.e., storing values in the strength of cortico-striatal synapses through DA-dependent plasticity)? If so, I think that the present model does not completely replace the traditional DA = TD-RPE hypothesis but instead replaces a part of it and complements the remaining part, and the authors' strong argument against the traditional hypothesis (in particular, the description mentioning "a paradigm shift" at the bottom of page 9) would need to be toned down.

v) The traditional DA = TD-RPE hypothesis does not only explain the response of DA neurons in association of sensory cue and primary reward but also explain the response of DA neurons or signals in DA-rich striatal/medial-prefrontal regions in more abstract value learning situations (e.g. with monetary reward or social reward) in longer time scales that may not correspond to time scales of synaptic plasticity (~ seconds). Can the FLEX model also explain such a wide range of observations?

Minor points:

- "However, additional data obtained more recently indicates this might not always be the case and that significant response to reward persists throughout training" in the middle of page 8. These can be (broadly) consistent with the DA = TD-RPE hypothesis (e.g., Gershman, 2014, Neural Comput; Morita & Kato, 2014, Front Neural Circuits).

- "by making ad-hoc modifications^{27,28}." at the bottom of page 8. I do not agree with the authors to describe these considerations as "ad hoc". In particular, distributional RL discussed in 27 is an important extension of RL theory even purely theoretically (separately from biological correspondence).

- "Synfire chains" at the top of page 11. I think this term is well known to computational neuroscientists but probably not to people in other fields, and so it may be good to explain it in a brief phrase. Also, it is good to add citation about the term "Synfire chains".

- "however in FLEX there is no analog for the value neurons assumed in TD " in the middle of page 18. Following this part, the authors argued that the FLEX model can be in line with some experimental findings suggesting computations without value representation. However, there are also a lot of experimental results (e.g., Samejima et al., 2005, Science) that indicate neural representation of values and seem consistent with the DA = TD-RPE hypothesis. How can the FLEX model be in line with these many results? (please see also major point iv) above)

(End)

Reviewer #3 (Remarks to the Author):

The standard model of temporal difference learning (TDL) assumes a hardwired set of basis functions to encode time. Here the authors point out that this component of TDL is probably not biologically plausible. That these hardwired basis function are implausible is often acknowledged, but I don't think specific models have been put forth in the context of TDL to explain how temporal basis functions could develop, as done in the current paper. Here a model in which the basis functions are learned is proposed based on the previous published model of timing from the lab. The model is placed in the context of TDL, including a feedback circuit between DA release which modulates synaptic plasticity and DA inhibition. An integrate-and-fire spiking model is developed, in which a population of neurons representing a cortical network undergoes DA modulated plasticity, and the "messenger" cells in the cortical network inhibit VTA DA neurons through GABA interneurons. First, the paper correctly highlights the problems with assuming a hardwired set of basis functions, and then proceeds to develop a model, and takes steps towards providing some experimental support for the model based on the reanalysis of published data. The proposal of a biologically plausible TDL model in regards the basis function is an important step forward, however as it currently stands it's a bit difficult to determine how consistent the model is with existing data.

The model deviates significantly from standard TDL predictions, and thus with a significant amount of published data. Thus it is important to determine which novel predictions are made and if they are consistent with the experimental data. The authors focus on the important prediction that total DA can transiently increase over the course of training. Support for this is provided by the reanalysis of published data (Fig 3 and S1), but this data perhaps, also shows a dip in total DA on Day 2. The authors seem to smooth over this a bit by averaging Day 1 and 2. I believe the model does clearly predict an increase in DA on day two which is not observed? Statistics should be performed per day not averaged in blocks over days, and the dip which may not be significant should be addressed, although the peak might not be significant either? The authors could probably also reanalyze other data sets including the Jeong et al data (?).

Does the model predict a discrete drop in US evoked DA as Figure 7 suggests? Additionally, the model seems to predict that the US evoked DA should remain constant until the temporal basis functions are learned. If these are indeed predictions are they consistent with the Amo or Jeong data?

The model predicts there is no backward migration of the DA response, but does predict a forward migration of the "messenger" neurons? Can this serve as an experimentally testable prediction? Either way it would helpful to list what the authors consider testable predictions that would

disprove the model.

There are some concerns with over weighing the results of the recent Jeong paper as an example that TDL is incorrect, because that paper might be taken to argue against the current model because they don't seem to see much of a characteristic DA dip during extinction trials, appearing to suggest there is no need for any temporal basis functions at all.

Minor comments.

To provide a more direct comparison with the experimental data, perhaps it would be best to not use the auROC but the more standard DA "firing rate" in Fig. 6 and 7.

The values of most of the parameters were not provided in the paper, and I don't think rho was defined.

Typo: cell responses that have evolve during trace conditioning.

Legend Figure 7A. State that it is total/integrated DA across the trial that is being plotted.

REVIEWER COMMENTS

Reviewer #1 (Remarks to the Author):

In this manuscript, Cone et al. propose a model of error-based learning in which the temporal basis is not fixed during the course of learning but is itself learned. They show that the resultant algorithm produces a signal related to mesolimbic DA responses that deviate from some RPE predictions. In this sense, they contrast their model with TD RPE. They show that a biophysically plausible implementation of their model can capture several experimental results related to dopamine.

Overall, I find that this work is valuable for the field, but has some limitations that dampen my enthusiasm. These are listed below.

1. Their core point here is that the temporal basis for associative learning need not be fixed. They show that a fixed basis can have some challenges. However, Hennig et al. showed recently that an RNN based model (“value RNN”) can learn cue-reward associations without being given an explicit state space. Indeed, they show that the resultant latent representations appear like commonly assumed state spaces after learning. However, these representations are themselves learned. Thus, this model demonstrates that TD can work without a fixed temporal basis. How do the authors compare their model to this alternative?

Thank you for comment. It is indeed important for us to explain how our work differs from previous work such as that by Henning et al (2023). The work by Hennig et al does not really use the TD algorithm, instead it uses the cost function used in TD for training an RNN, by using a BPTT algorithm. Such algorithms are based on the calculation of a gradient, and in BPTT this approach breaks both locality and causality and is therefore not biophysically plausible. Such an approach overcomes the problems of biophysically implausible assumptions about the network structure by using a powerful but biophysically implausible learning rule. Our analysis and approach is centered on biophysically plausible solutions. In the revised version we have tried to clarify this point, and explain the distinction between our approach and approaches based on gradient based algorithms. (see pg 7)

2. A core point that the authors make is that when distractor stimuli are presented during a cue-reward simulation, an RNN will not develop a fixed basis set. As above, Hennig et al’s RNN model does not have a fixed basis set but still learns the task. So, I am not sure that this is a convincing argument. Further, it is not clear to me that FLEX will naturally solve this task. Can the authors show that their model will learn the task shown in Figure 2 and naturally identify the distractors as being irrelevant?

There are two points we can infer from this comment, one that the Hennig model can learn even with distractors and the second to doubt if FLEX can learn with distractors. To address the second point, we have added a new supplemental figure (Fig. S5) and text (pg 16) to show that FLEX can learn with distractors. In addition, it is unclear to us how the Hennig approach would respond to distractors. If the task was learned without distractors, but then distractors were presented, which affect the dynamics of the RNN, it seems to us that the performance of the network would be impaired. In contrast, it might be possible to teach the network to ignore specific distractors during the learning phase. Because we do not know to what extent the Hennig approach can address distractors we have not elaborated on this point in the paper.

3. How well does FLEX deal with long cue-reward delays such as those shown in experimental results (e.g., up to 16 s in some Schultz papers)?

This is a very reasonable question. Learning activity on long timescales (orders of magnitudes larger than the neural time constant) is generally a bottleneck in recurrent networks, particularly when considering biophysically plausible learning rules (as we do in this work).

There are two sources for this bottleneck, one stems from intrinsic limitations of biophysically plausible spiking RNNs and another stemming from the time scale of eligibility traces in the learning rule.

We have previously shown (Aksoy et al 2022) that with additional biophysically plausible single cell mechanisms stochastic spiking RNN's can be trained for slowly decaying activity lasting up to tens of seconds, as observed experimentally in this system. We have not used these additional mechanisms here, but now relate to this problem and cite our previous work as a partial solution.

Behaviorally, cue reward associations can be learned for gaps on the order of minutes, although the physiological mechanisms are less well explored. A mechanism in which the temporal basis is constructed by a spiking noisy recurrent network is unlikely to be the basis of learning associations with delays on the order of minutes. For such long durations, a different class of models would be required.

As stated, FLEX is a general theory. For gaps between stimulus and reward that span minutes, our current model is insufficient, and we are also not aware of data from DA neurons at these time scales. To account for longer durations, other specific biophysical models would be required though they could also fall into the general FLEX framework. We have added a discussion of these issues in the current version of the paper (pg. 19).

4. While I am familiar with many similar claims of mismatch of RPE and experimental data, I am not familiar with the integral being less than 1. Can the authors mathematically demonstrate that TD-RPE must have an integral less than 1 for a Markov state space?

Thank you for your comment, we have amended our supplemental material to include evidence that the integral of RPE should be less than or equal to 1. For the TD(0) case we provide an analytical proof, and we supplement it by simulations which demonstrate that this result holds for more complex versions of TD.

5. As best as I understand, FLEX is a model of learning associations and not value. A fundamental property of dopamine responses is scaling of cue responses by predicted reward magnitude. Can FLEX account for this?

While the model was not originally designed to encode the reward magnitude directly, as our aim was to show that a temporal basis can be learned, we have added a supplemental figure (Fig. S6) and a text in the paper (pg 17) demonstrating that in a slightly modified version of the model in which cue responses are indeed scaled by predicted reward magnitude. This alternate model assumes that some signal of reward magnitude still exists at the time of US. This could be achieved, for example, by other neuromodulators are released alongside dopamine upon reward, and when rewards are predicted, dopamine release is suppressed while the release of these other

neuromodulators is not. This allows for the eligibility traces associated with cue learning to bind to these neuromodulatory signals even after the reward has been predicted (and dopamine suppressed), which allows for an encoding of reward magnitude by the cue both during and after training.

Reviewer #2 (Remarks to the Author):

The prevailing hypothesis that dopamine (DA) represents temporal-difference (TD) reward prediction error (RPE) has been supported by many experimental results, but there are also a number of findings that appear inconsistent with the hypothesis, including that temporal shift of DA response from reward timing to cue timing (predicted from the hypothesis) is not always observed. Also, the DA = TD-RPE hypothesis presumes that each sensory cue stimulus evokes, by default, a specific temporal basis function, on which TD-RPE-based value learning occurs, but this presumption itself has a theoretical difficulty (it is practically impossible to generate basis function for "every" stimulus, but it is also impossible to generate it only for "relevant" stimulus without a priori knowledge about which stimulus is relevant). In order to address these difficulties in the traditional DA = TD-RPE hypothesis, the authors developed a new neurocomputational model, the FLEX model, that explains how such a stimulus-specific temporal basis function for reward-associated stimulus, initially not existing, can be gradually shaped through plasticity mechanisms. The authors showed that the model can be in line with experimental results that appear at odds with the DA = TD-RPE hypothesis.

I would agree with the authors that the issue tackled by this work is a central question in neuroscience or even widely in related sciences. I also like the FLEX model, its potential explanatory power and also its uniqueness as a model bridging the algorithmic level (TD reinforcement learning (RL)) and the implementation level (neuronal plasticity). However, I think there remain major points, described below, that are desired to be addressed or clarified.

i) I expected that the FLEX model was first introduced at the algorithmic level (Marr's 2nd level), but the authors almost immediately went to the implementation level (Marr's 3rd level). After reading the whole article, I gradually understood that in the case of the FLEX model, these two levels are really integrated together, and it may be difficult to describe its algorithm, as a set of equations, separately from implementation. But still, it would be helpful if the core mechanism of the model is first described before going to implementation. In the introduction, the authors explained when there are many sensory stimuli, A, B, C, etc, and only some of them, e.g., C, is associated with reward after a certain delay (unknown to agent) while other stimuli are just noise, the C-reward association is difficult to be learnt by fixed temporal basis function(or attractor dynamics) starting from each stimulus or general temporal basis function. These explanations are very nice, but what is lacking, I think, is a conceptual explanation of what the FLEX model is and how it can resolve this critical issue. In fact, there is already a corresponding description (section "The FLEX theory of reinforcement learning: A theoretical framework based on a plastic temporal basis.", page 10-11), but I think it is too broad and does not point to how the FLEX model resolves the critical issue (distinguishing C from other stimuli).

The Marr three level framework is useful conceptually and it also aids in explaining theories. We are not sure how to describe an abstract algorithm for FLEX (Marr level 2) separately from the algorithm's implementation (Marr level 3). However, we have chosen to describe separately the

general FLEX theory, and our specific mechanistic implementation. The FLEX theory simply states that any plausible algorithm must be able to flexibly generate a temporal basis for the salient cues. One problem which makes it hard to describe a general algorithm for FLEX is that the FLEX theory does not specify precisely the nature of the temporal basis, our specific model generates a useful basis, but is clearly not the only option. We have not described a general algorithm that can do this, but instead presented a biophysically plausible implementation as a proof of concept. Indeed, by doing this we have skipped over the level 2 description.

ii) Related to point i) above, I think it would be necessary to show the abovementioned point (i.e., activity starting from C is formed while activity starting from A or B is not) by actual simulation. Currently, the authors first described the case where there is only a single sensory stimulus (page 13-), and then described the case where there are sequentially presented multiple (high-order) cue stimuli (page 16-). But this latter case differs from the situation where there are many irrelevant stimuli (A, B, ..) and only some (C) is relevant, because all the sequentially presented stimuli are relevant (reward-associated) ones. I would like to see actual simulation results on how the FLEX model realizes formation of cue-specific sustained activation (temporal basis function) only for relevant stimuli.

We agree that it is important to show this behavior directly by simulation, we have amended our supplemental material to include a figure (supplementary figure S5 which demonstrates that distractor cues do not affect our network's timing (described in pg 16)).

iii) Regarding the point that cue-specific sustained activation (temporal basis function) is shaped through plasticity mechanisms, there exists a previous study that is not cited in the present manuscript:

Nicolas Deperrois, Victoria Moiseeva, & Boris Gutkin

Minimal Circuit Model of Reward Prediction Error Computations and Effects of Nicotinic Modulations

Front. Neural Circuits, Volume 12 - 2018

<https://doi.org/10.3389/fncir.2018.00116>

I think it would be necessary to cite this work and discuss how the FLEX model is conceptually different from (and/or superior to) this previous model (or if the authors think discussing this previous work is not necessary, please explain the reason). I think this point is especially important given that the authors claim that the FLEX model is a "proof-of-concept" model; how does it significantly depart from the concept of the previous work by Deperrois et al.?

We greatly appreciate this comment as we were not aware of this paper previously, and did not find it in our searches while working on this paper. This is excellent work, which is similar to ours in many respects. Although there are specific differences in the details, we see this model as another implementation of a FLEX model.

The details of the Deperrois et al (2018) model differ from ours in several key respects. First, their PFC model has a slow time constant of 1000ms introduced by the adaptation variable, which is responsible for the slow time scale of decay in this, model which is different from our PFC component which depends on an RNN without this postulated slow component. Second, the PFC module in the Deperrois paper does not have the equivalent of our messenger cells, consequently it

could not account for VTA response for a reward that appears earlier than expected, as in the data and our model. Third, the learning rules used in that paper are quite different, they assume that the recurrent synapses within PFC have access to a variable representing the time between reward, and the network decay. They also assume that the cortical striatal synapses have access to the difference between the predicted response at CS to the magnitude of the US. Note that these two quantities are separated in time. Making this assumption simplifies the learning of CS magnitude, but does not seem biophysically plausible. We do not make these assumptions. In our model we use biophysically plausible learning rules based on eligibility traces, for which there is experimental evidence. Fourth, the DePerrier paper used a deterministic rate based MFT model, we use a model with stochastic spiking neurons which enable us to also explore the variability between single cell responses, as seen in the data. Another difference is that our model does not take into account the role of Ach which is extensively studied in the DePerrier model. In addition to the difference in the model itself, we also simulate a somewhat different set of experiments than they do.

More importantly, apart from these specific differences, much of our current work concentrates on demonstrating that the previously used TD approach is not biophysically reasonable and is inconsistent with experimental data. However, because we are now aware of this paper we discuss it, describe it in the context of the FLEX theory, and briefly explain how it differs from our model.

iv) The authors' (FLEX model's) claim that DA does not represent TD-RPE by default but such a representation is developed through plasticity mechanisms is certainly an important departure from the traditional view. However, once DA's representation of TD-RPE is learnt, can it be used to implement TD RL (i.e., storing values in the strength of cortico-striatal synapses through DA-dependent plasticity)? If so, I think that the present model does not completely replace the traditional DA = TD-RPE hypothesis but instead replaces a part of it and complements the remaining part, and the authors' strong argument against the traditional hypothesis (in particular, the description mentioning "a paradigm shift" at the bottom of page 9) would need to be toned down.

It is indeed possible that the temporal basis developed by FLEX could subsequently be used by a traditional TD algorithm. Such an assumption might leave a role for traditional TDRL after initial learning phase in which the basis functions are learned. However, the DA release in our model is testably different from an RPE (Figure 7), so it does directly challenge the DA=TD-RPE hypothesis. Despite this, we agree that the language regarding a "paradigm shift" is too strong, and as such, we have removed it.

v) The traditional DA = TD-RPE hypothesis does not only explain the response of DA neurons in association of sensory cue and primary reward but also explain the response of DA neurons or signals in DA-rich striatal/medial-prefrontal regions in more abstract value learning situations (e.g. with monetary reward or social reward) in longer time scales that may not correspond to time scales of synaptic plasticity (~ seconds). Can the FLEX model also explain such a wide range of observations?

The power of the traditional TD formulation is its normative basis, which can be used for more complex and abstract value learning scenarios. The FLEX formulation is primarily concerned with biophysically plausible mechanisms. When FLEX converges its response properties are similar to those of the traditional TD formulation, but not during the learning process. It might be seen as a

mechanistic implementation that approximates TD in some cases. We have also shown the response of FLEX in a slightly more complex paradigm of sequential conditioning. We have shown results that are consistent with experimental results and may even account for apparent contradictions in previous results. We have not tested whether FLEX can account for more abstract and complex value estimates and over much longer time scales. Because of the normative definition of TD is simpler to apply it to these more complex scenarios. In contrast, because mechanistic models of FLEX require a plausible mechanism, applying FLEX to more abstract paradigms might be less straightforward. It is quite feasible that additional mechanisms would be required to account for these cases. For example, as noted above for processes with much longer time scales.

Minor points:

- "However, additional data obtained more recently indicates this might not always be the case and that significant response to reward persists throughout training" in the middle of page 8. These can be (broadly) consistent with the DA = TD-RPE hypothesis (e.g., Gershman, 2014, *Neural Comput*; Morita & Kato, 2014, *Front Neural Circuits*).

While these references address spatial navigation tasks in which there is ramping as the target is approached., our model is concerned with the simpler trace conditioning tasks in which the response at the time of the US does not completely go away, but there is no ramp leading to it. We have modified the sentence and added references to the type of persistent response we had in mind: "However, additional data obtained more recently indicates this might not always be the case and that significant response to reward **at the time of the actual reward persists** throughout training (citation numbers)."

- "by making ad-hoc modifications^{27,28}." at the bottom of page 8. I do not agree with the authors to describe these considerations as "ad hoc". In particular, distributional RL discussed in 27 is an important extension of RL theory even purely theoretically (separately from biological correspondence).

We have changed the language we use here. We no longer say ad-hoc. We now say "by making significant modifications to the classical value-based formulation of TD". That this is a significant change from classical TD is clearly stated in the papers cited. The following is taken from Dabney et al. 2020: "In contrast to classical temporal-difference (TD) learning, distributional RL posits a diverse set of RPE channels, each of which carries a different value prediction, with varying degrees of optimism across channels. (Value is formally defined in RL as the mean of future outcomes, but here we relax this definition to include predictions about future outcomes that are not necessarily the mean.)"

- "Synfire chains" at the top of page 11. I think this term is well known to computational neuroscientists but probably not to people in other fields, and so it may be good to explain it in a brief phrase. Also, it is good to add citation about the term "Synfire chains".

Thank you for pointing this out – we have added references for readers who may not be familiar with the term.

- "however in FLEX there is no analog for the value neurons assumed in TD " in the middle of page 18. Following this part, the authors argued that the FLEX model can be in line with some experimental findings suggesting computations without value representation. However, there are also a lot of experimental results (e.g., Samejima et al., 2005, Science) that indicate neural representation of values and seem consistent with the DA = TD-RPE hypothesis. How can the FLEX model be in line with these many results? (please see also major point iv) above)

This claim is a statement of fact about FLEX, the model does not have explicit value neurons. Whether the brain has value neurons (not RPE neurons) is contentious although some labs have reported them others claim their absence (see Hayden and Niv, 2021). This paper does not make the claim that value neurons are not or should not be found in the cortex, it simply states that we do not need them in order to account for the plasticity if DA neurons in VTA. We have now tried to clarify this better in the text.

Reviewer #3 (Remarks to the Author):

The standard model of temporal difference learning (TDL) assumes a hardwired set of basis functions to encode time. Here the authors point out that this component of TDL is probably not biologically plausible. That these hardwired basis function are implausible is often acknowledged, but I don't think specific models have been put forth in the context of TDL to explain how temporal basis functions could develop, as done in the current paper. Here a model in which the basis functions are learned is proposed based on the previous published model of timing from the lab. The model is placed in the context of TDL, including a feedback circuit between DA release which modulates synaptic plasticity and DA inhibition. An integrate-and-fire spiking model is developed, in which a population of neurons representing a cortical network undergoes DA modulated plasticity, and the "messenger" cells in the cortical network inhibit VTA DA neurons through GABA interneurons. First, the paper correctly highlights the problems with assuming a hardwired set of basis functions, and then proceeds to develop a model, and takes steps towards providing some experimental support for the model based on the reanalysis of published data. The proposal of a biologically plausible TDL model in regards the basis function is an important step forward, however as it currently stands it's a bit difficult to determine how consistent the model is with existing data.

The model deviates significantly from standard TDL predictions, and thus with a significant amount of published data. Thus it is important to determine which novel predictions are made and if they are consistent with the experimental data. The authors focus on the important prediction that total DA can transiently increase over the course of training. Support for this is provided by the reanalysis of published data (Fig 3 and S1), but this data perhaps, also shows a dip in total DA on Day 2. The authors seem to smooth over this a bit by averaging Day 1 and 2. I believe the model does clearly predict an increase in DA on day two which is not observed? Statistics should be performed per day not averaged in blocks over days, and the dip which may not be significant should be addressed, although the peak might not be significant either? The authors could probably also reanalyze other data sets including the Jeong et al data (?).

Thanks for these comments. Indeed, it would be better to have much more data of DA neuronal responses during training in order to make our point. The main reason for showing that in experimental data the total integral over DA is not monotonically decreasing is to demonstrate that existing data is inconsistent with classical TD models. Indeed, in our implementation the dynamics of

the DA integral is similar to the data we have shown, but we see this as secondary reason for showing the data because even in our model the exact DA dynamics are parameter dependent and in other implementations of FLEX they might be quite different.

In the Amo et al data we group two days together to obtain statistical significance, otherwise the statistical tests lack sufficient power. This analysis is therefore restricted by the available data. At the reviewers suggestion we have also analyzed the data from Jeong et. al, and from Coddington and Dudman. In both those data sets the integral shows a similar trend. In the Jeong et al data it is not statistically significant, in the Dudman and Coddington it is. In the analysis of the Coddington and Dudman data the statistics were obtained by comparing non-overlapping groups of 10 data points with the first 10 data points, thus no averaging over days was needed. For the Amo et al data we had to average over days in order to obtain statistical significance, otherwise the n was too small to obtain statistical significance. We have now used Dudman and Coddington instead of the Amo et al data in figure 3 and have moved the Amo et al data to supplemental figure 1.

We have attached below a figure of the Jeong et al data. The black line shows the average trend but it's not statistically significant. The lack of significance stems here from a large animal by animal variability combined with a small n=7. We estimate that with similar variability, an n=12 at least will be required for statistical significance. Due to the lack of statistical significance, we have not added these results as a supplemental figure

Figure above: Analysis of Jeong data. Colored lines for individual animals, black line average over animals.

Does the model predict a discrete drop in US evoked DA as Figure 7 suggests? Additionally, the model seems to predict that the US evoked DA should remain constant until the temporal basis functions are learned. If these are indeed predictions, are they consistent with the Amo or Jeong data?

We are not entirely clear to what “discrete drop” you refer – we will assume that it is the quick transition in Figure 7a at around trial 100, in which the US-evoked DA is quickly suppressed. The sharpness of the drop is dependent on the learning rate of the M->VTA weights. The model is robust to a reduction in this learning rate, however, we leave it high to illustrate the point of Figure 7, that in our model DA need not “migrate” from the time of the US to the time of the CS in an equal and

opposite fashion. The model does indeed predict that the US evoked DA should remain constant until the temporal basis functions are learned, however, this is not tested in either the Amo or Jeong data (no putative temporal basis functions, say from PFC, are recorded). Further experiments are needed to test this hypothesis.

The model predicts there is no backward migration of the DA response, but does predict a forward migration of the “messenger” neurons? Can this serve as an experimentally testable prediction? Either way it would be helpful to list what the authors consider testable predictions that would disprove the model.

Yes, the model would predict a forward migration of the timer and messenger neurons. We have included this, alongside other predictions, in additional text in the discussion.

There are some concerns with overweighing the results of the recent Jeong paper as an example that TDL is incorrect, because that paper might be taken to argue against the current model because they don't seem to see much of a characteristic DA dip during extinction trials, appearing to suggest there is no need for any temporal basis functions at all.

The Jeong et al paper argues against the normative assumptions of TD. It argues that the DA signal at time of CS is at least partially retrospective rather than prospective, and we have discussed and referenced it for completeness. There is ample data indicating a reduction in the response magnitude at the time of US, this though might require extensive training. This is even apparent in the Jeong et al data (Fig. 4C), though we have not tested for its significance. Additionally, in some of the published data (e.g: Cohen et al 2012) one observes heterogeneity in the reduction of DA at the time of US, with some neurons being completely inhibited while others are not affected at all. This heterogeneity is consistent with our model (Fig 6d). We have now tried to clarify the subtle distinction in the new version that while the Jeong et al data does argue against the normative basis of TD, their data has no bearing on the mechanism of the plasticity of dopamine release.

Minor comments.

To provide a more direct comparison with the experimental data, perhaps it would be best to not use the auROC but the more standard DA “firing rate” in Fig. 6 and 7.

We actually agree that firing rates make more sense, and that auROC is an additional confusing and unnecessary processing stage. However, since auROC was used in the experiment in which we directly compare our Fig 6 and 7 results to (Cohen et al 2012), we felt that its usage was appropriate. If the reviewer insists we could easily change this, but it would no longer be a direct comparison to Cohen et al 2012.

The values of most of the parameters were not provided in the paper, and I don't think rho was defined.

Thanks for the comment, we've added a parameter table to the supplemental material, and made sure all parameters are defined in the methods.

Typo: cell responses that have evolve during trace conditioning.

Fixed

Legend Figure 7A. State that it is total/integrated DA across the trial that is being plotted.

Fixed

Reviewer #1 (Remarks to the Author):

The authors have done a good job addressing my concerns. I am okay with publication of the manuscript. However, I have a few more comments they might wish to consider. I do not need to see the revision again.

In the Appendix, there is an alpha missing in A-3 and an I missing in Δ_{n-1} in A-4. The assumptions for this derivation seem a bit limiting to me. Is the core idea that the average RPE across all states should reduce over learning since it is the reduced objective function during value learning? I guess sum of RPE across all states weighted by frequency of state occurrence should be zero if the value function is correctly learned.

Reviewer #2 (Remarks to the Author):

The authors have addressed most of the points that I raised originally. There remains one, but still important, point:

In reply to my comment and also Reviewer 1's similar comment, the authors have now shown that FLEX works fine in the presence of distractor *during recall* (Supplemental Figure 5). However, what I intended to ask was whether FLEX can still work if distractors (irrelevant stimuli) exist *during learning*.

Regarding how the authors' work relates to the previous work by Deperrois et al. (now cited as [69]), I understand the authors' explanation. There is only a type remained:

This previous publication different from our model in the details of the recurrent network...

-> This previous publication differs from our model...?

Reviewer #3 (Remarks to the Author):

The authors have done a good job addressing my concerns. As I stated in my first review I believe this is first presentation of network model of the learned encoding of timed responses in the context of TDR, where there has been an assumption of hardwired temporal basis functions. As such I think the paper is a significant contribution to the field. I also agree with the authors that the Deperrois et al, 2018 paper is fundamentally different as it relies on hard-wired intrinsic time constants rather than emergent network properties.

While the presented model is unique in that it frames the emergence of temporal basis function in the context of RL, the authors acknowledge models that have shown how temporal basis function can emerge through supervised and unsupervised mechanisms in RNNs, (e.g., Liu & Buonomano, 2009, Fiete et al, 2010).

On a minor note: although I understand the term synfire chain is often used to refer to a neural sequence, the term synfire chain as originally coined by Abeles actually refers to synchronous feedforward (thus the prefix syn) chains and does not really apply to neural sequences in typical models of timing. I would recommend sticking to the term "neural sequences".

REVIEWER COMMENTS

Reviewer #1 (Remarks to the Author):

The authors have done a good job addressing my concerns. I am okay with publication of the manuscript. However, I have a few more comments they might wish to consider. I do not need to see the revision again.

In the Appendix, there is an alpha missing in A-3 and an I missing in $I_{\Delta_{n-1}}$ in A-4. The assumptions for this derivation seem a bit limiting to me. Is the core idea that the average RPE across all states should reduce over learning since it is the reduced objective function during value learning? I guess sum of RPE across all states weighted by frequency of state occurrence should be zero if the value function is correctly learned.

Thank you for comment. We are grateful that you have spotted the typos in the Appendix. Regarding the derivation in the appendix, its assumptions mean that this calculation applies only to TD(0), and for either a tabular temporal basis or a delta function temporal basis. Using simulations, we show that the conclusion, that the integral is non-increasing, holds for TD(λ) as well, though the temporal dynamics of the decrease are more rapid. As the value function is learned the integral decreases, and the level of decrease depends on the parameter γ . If $\gamma=1$, the integral does not decrease. However, the sum of all responses does not go to zero. When learning converges, for these basis functions, there is only a response (an RPE) to the initial state, because that state triggered by the CS, is unpredictable. The RPE for all other states is indeed zero after convergence. After convergence, the integral is equivalent to the RPE in the initial state.

Reviewer #2 (Remarks to the Author):

The authors have addressed most of the points that I raised originally. There remains one, but still important, point:

In reply to my comment and also Reviewer 1's similar comment, the authors have now shown that FLEX works fine in the presence of distractor *during recall* (Supplemental Figure 5). However, what I intended to ask was whether FLEX can still work if distractors (irrelevant stimuli) exist *during learning*.

Regarding how the authors' work relates to the previous work by Deperrois et al. (now cited as [69]), I understand the authors' explanation. There is only a type remained:

This previous publication different from our model in the details of the recurrent network...

-> This previous publication differs from our model...?

Thank you for following up, we poorly worded our reference to that figure – the distractor cues in Supplemental Figure 5 are indeed intermittently presented during learning (not associated with a reward). We show the results from after learning, to show that their presence does not disrupt the learned timing of the stimulus C. We have added clarification to the figure caption, and revised the section in the text to highlight that the cues are shown during learning, and the network is invariant to their presence after learning.

We have also fixed the typo, thank you for pointing it out.

Reviewer #3 (Remarks to the Author):

The authors have done a good job addressing my concerns. As I stated in my first review I believe this is first presentation of network model of the learned encoding of timed responses in the context of TDR, where there has been an assumption of hardwired temporal basis functions. As such I think the paper is a significant contribution to the field. I also agree with the authors that the Depperois et al, 2018 paper is fundamentally different as it relies on hard-wired intrinsic time constants rather than emergent network properties.

While the presented model is unique in that it frames the emergence of temporal basis function in the context of RL, the authors acknowledge models that have shown how temporal basis function can emerge through supervised and unsupervised mechanisms in RNNs, (e.g., Liu & Buonomano, 2009, Fiete et al, 2010).

On a minor note: although I understand the term synfire chain is often used to refer to a neural sequence, the term synfire chain as originally coined by Abeles actually refers to synchronous feedforward (thus the prefix syn) chains and does not really apply to neural sequences in typical models of timing. I would recommend sticking to the term “neural sequences”.

Thank you for review.

This is a good point regarding the use of “synfire chains”, we have changed all references to “sequential neural sequences” or similar.